



# Can we obtain consistent estimates of the emissions in Europe from three different CH$_4$ TROPOMI products?

Aurélien Sicsik-Paré[1], Audrey Fortems-Cheiney[1,*], Isabelle Pison[1], Grégoire Broquet[1], Alvin Opler[1], Elise Potier[1,*], Adrien Martinez[1], Oliver Schneising[2], Michael Buchwitz[2], Joannes D. Maasakkers[3], Tobias Borsdorff[3], and Antoine Berchet[1]

[1]Laboratoire des Sciences du Climat et de l'Environnement, LSCE/IPSL, CEA-CNRS-UVSQ, Université Paris-Saclay, F-91191 Gif-sur-Yvette, France
[*]now in Science Partners, Quai de Jemmapes, 75010 Paris, France
[2]Institute of Environmental Physics (IUP), University of Bremen FB1, Bremen, Germany
[3]SRON Netherlands Institute for Space Research, 2333 CA Leiden, The Netherlands

**Correspondence:** Aurélien Sicsik-Paré (aurelien.sicsik-pare@lsce.ipsl.fr)

**Abstract.** Satellite observations from the Sentinel-5P TROPOMI instrument, combined with inverse modeling, provide a valuable resource for quantifying regional methane (CH$_4$) emissions. This study compares the emissions estimated from variational inversions in 2019 over Europe (0.5° resolution) assimilating three TROPOMI products of dry-column methane mole fractions (XCH$_4$). The SRON (v2.4, operational product), BLENDED (v1.0), and WFMD (v1.8) products are retrieved from distinct

algorithms. They differ in coverage, error characterization, and XCH$_4$ spatial distribution. Results indicate that the largest contributions to XCH$_4$ differences may be attributed to aerosol scattering and sensitivity to albedo. The derived 2019 European CH$_4$ emission budgets show a relative increase of +2% for SRON, and a decrease of -1%, -33% and -9%, respectively, for BLENDED, WFMD and surface-based inversions. Seasonal emissions are highly correlated across the inversions. Spatial emission patterns and optimized boundary conditions are similar for the non-independent SRON and BLENDED but differ

substantially from WFMD. Evaluation with independent surface stations shows error reduction for about half of the sites, with BLENDED performing best. However, no product is systematically closer to the emissions estimated when assimilating surface observations. Observing System Simulation Experiments (OSSEs) are used to disentangle the drivers of differences between the posterior emissions. They reveal that observation density and errors, but also averaging kernels and prior profiles play a key role in the inversion's capacity to constrain the emissions. Using consistent error definition and quality filters increases the

consistency of the OSSEs, paving the way for more consistent emission estimates.



## 1 Introduction

The global emission pathways to limit global warming below the international objective of 1.5°C (IPCC, 2021) include significant reductions of methane ($CH_4$) emissions. Methane is the second most important anthropogenic greenhouse gas (GHG) after carbon dioxide ($CO_2$). Despite its relatively short atmospheric lifetime of $9.1 \pm 0.9$ years (Szopa et al., 2023), mainly due

to oxidation by OH radicals in the troposphere, it has a strong radiative efficiency: its Global Warming Potential is ∼81 times that of $CO_2$ over 20 years and ∼28 times over 100 years (Forster et al., 2021). Global atmospheric concentrations of $CH_4$ have increased by about 262% since the pre-industrial period, with higher rates over the last 20 years (Lan et al., 2022). An accurate accounting of emissions is required to assess the effectiveness of current regulatory policies and to provide a robust reference for projections.

$CH_4$ emissions reporting is currently done by most countries with bottom-up (BU) approaches which include inventories following the methodological procedures from the Intergovernmental Panel on Climate Change (IPCC) Guidelines (IPCC, 2006, 2019), and biogeochemical models. However, large uncertainties still affect the quantification of $CH_4$ emissions (Saunois et al., 2020; Solazzo et al., 2021; Petrescu et al., 2024): the yearly budget of $CH_4$ emissions in Europe is approximately 50 Tg/yr between 2010 and 2019 according to Saunois et al. (2020), with an uncertainty estimated to 40%.

Inverse modeling of $CH_4$ emissions combining atmospheric observations and transport simulations is an important top-down (TD) tool for complementing BU approaches by using independent information provided by atmospheric mixing ratios to reduce uncertainties on the emissions. In the last decade, satellite data of $CH_4$ total column dry air mixing ratios ($XCH_4$) (Monteil et al., 2013; Cressot et al., 2014; Alexe et al., 2015; Bergamaschi et al., 2015) have provided wider spatial coverage than the surface data but with a double challenge: the estimation of the $XCH_4$ retrieval from the raw spectroscopic measurement, and

the estimation of emissions based on the assimilation of retrievals. Regarding methane, the Greenhouse Gases Observing Satellite (GOSAT) was launched in 2009 and provides relatively accurate but sparse $XCH_4$ observations (Parker et al., 2020). The TROPOspheric Monitoring Instrument, also known as TROPOMI (Veefkind et al., 2012), launched in October 2017 on-board the satellite Sentinel 5-P, now provides $XCH_4$ with a nadir resolution of 5.5 km × 7 km and daily global coverage (Hu et al., 2016; Hasekamp et al., 2022). Because of its high resolution (relatively to previous instruments) together with its high cov-

erage, TROPOMI retrievals have been successfully used to detect large releases from oil and gas facilities (Schneising et al., 2020; Zhang et al., 2020; Lauvaux et al., 2022; Veefkind et al., 2023), as well as coal mines and landfills (Schuit et al., 2023). They have also been used to quantify national and sectoral emissions, in global (Qu et al., 2021; Yu et al., 2023) and regional inversions for the US (Shen et al., 2022; Nesser et al., 2024), the Middle East and North Africa (Chen et al., 2023), East Asia (Chen et al., 2022; Liang et al., 2023) and South America (Nathan et al., 2023; Hancock et al., 2024).

The TROPOMI $XCH_4$ total columns are retrieved from radiance measurements in the 2.3 $\mu$m SWIR spectral range, along with their corresponding averaging kernels (AKs), which assess the sensitivity of the retrievals to the different atmospheric layers. This derivation is complex and remains a major challenge. The retrieved columns and the vertical sensitivity indeed strongly depend on different factors of the radiative transfer (e.g., clouds, the assumed prior profile shape, surface albedo, strato-



spheric background, aerosols...), as well as on the algorithm used for the retrieval. The systematic analysis of the TROPOMI
XCH$_4$ data has pointed to biases linked to albedo and scattering issues (Barré et al., 2021).

In this context, several products of TROPOMI XCH$_4$ retrievals have been developed with different radiative transfer inverse
modeling algorithms, and they are routinely updated. The SRON Netherlands Institute for Space Research provides the oper-
ational Copernicus product (European Space Agency, 2021), which has recently been updated with the optimized settings of
the beta research product (Lorente et al., 2021, 2023). A destriping algorithm is also applied to new XCH$_4$ data (Borsdorff
et al., 2024). It uses the full-physics algorithm RemoTeC and simultaneously retrieves XCH$_4$, surface albedo and atmospheric
scattering properties. The BLENDED TROPOMI+GOSAT product (Balasus et al., 2023) is a corrected version of the SRON
operational product. It relies on a machine learning model, which was trained to predict the differences between TROPOMI
and GOSAT co-located retrievals. The correction was then applied to the whole SRON dataset. The scientific WFMD product
from University of Bremen is independent from the two previous ones and based on the retrieval algorithm Weighting Function
Modified Differential Optical Absorption Spectroscopy (WFMD-DOAS) (Schneising et al., 2019, 2023).

Previous inter-comparisons only include the SRON and WFMD products (T. Hilbig et al., 2023). A comparison of obser-
vations in the northern high latitudes (Lindqvist et al., 2024) revealed higher XCH$_4$ in the WFMD product in comparison to
SRON, as well as a persistent seasonal bias for the SRON operational product (high values in spring, low values in autumn). The
assimilation of older versions of SRON and WFMD products at high latitudes showed similarities in the posterior emissions
in terms of spatial distribution and time series but also some clear differences (Tsuruta et al., 2023). Despite the advancements
in XCH$_4$ retrievals in recent updates and the additional product BLENDED, the literature on systematic comparisons between
the different products and the consistency and applicability of these products for inversions remains limited.

This study aims at addressing these limitations by providing a comparison of methane emissions inferred from the assim-
ilation of the SRON, BLENDED and WFMD products in regional inversions, thereby exploring the robustness of inversions
with TROPOMI XCH$_4$ observations. The objective is to assess the consistency between retrievals of the three datasets and the
implication of the possible inconsistencies on resulting emission estimates. This benchmark requires the characterization of the
(in)consistencies in the observation coverage, observation errors and spatial biases of XCH$_4$. The study focuses on Europe in
2019, where an extensive network of surface stations provide independent in-situ CH$_4$ measurements that are used to support
the comparison.

First, we compare the three products and their simulated equivalents over Europe. We investigate the role of atmospheric,
instrumental and algorithmic variables in driving the XCH$_4$ differences between products. Using a machine learning model,
feature contributions to the prediction of the differences are quantified, following the methodology of Balasus et al. (2023).

Subsequently, we perform variational inversions to estimate the CH$_4$ fluxes at a $0.5° \times 0.5°$ spatial resolution and weekly tem-
poral resolution by assimilating each product independently. We conduct Observing System Simulation Experiments (OSSEs)
with synthetic pseudo-observations and perturbed prior fluxes. The OSSEs assess the ability of the inversion system to refine
emission estimates and its sensitivity to parameters such as observation density, error characteristics, and inter-product dif-
ferences. Scenarios isolating specific changes provide insights into the drivers of differences in the posterior fluxes. Finally,
we compare European CH$_4$ emission estimates in 2019 from the three TROPOMI products and ground-based observations, at



the pixel, country, and regional resolution. For this purpose, we use the recent inverse modeling platform Community Inversion Framework (CIF Berchet et al., 2021), coupled with the regional Eulerian atmospheric chemistry-transport model (CTM) CHIMERE (Menut et al., 2013; Mailler et al., 2017) and its adjoint code (Fortems-Cheiney et al., 2021). CHIMERE has already been used to model transport of GHGs at the regional scale, especially $CO_2$ (Broquet et al., 2011; Santaren et al., 2021) and $CH_4$ (Pison et al., 2018). The methane observations, the configuration of the CHIMERE CTM and the methodology for the variational inversions and OSSEs are described in Section 2. The results, particularly those from the inversions and the OSSEs, are detailed in Section 3.



## 2 Data and Methods

The purpose of our atmospheric inversions is to correct prior estimates of the $CH_4$ emissions, to improve the fit between the simulations of $CH_4$ atmospheric mixing ratios provided by a CTM and the observations. In this section, we detail the inputs and components of the CIF-CHIMERE inversion system. Section 2.1 provides an overview and a comparison of the coverage, distribution and errors of the three TROPOMI products. It also introduces the other $CH_4$ observations used in this study. Section 2.2 describes the prior emission estimates, and Section 2.3 details the configuration of the CHIMERE CTM. The methodology for the inversions and OSSEs is outlined in Section 2.4.

### 2.1 $CH_4$ observations

#### 2.1.1 TROPOMI satellite products

The TROPOspheric Monitoring Instrument was launched onboard Sentinel 5 Precursor (S5P) in 2017. This satellite operates on a sun-synchronous orbit with a Local Time of Ascending Node of 13:30 and a reference altitude of 824 km. The instrument is an imaging spectrometer that measures in the UV-visible (270 – 500 nm), the near infrared (NIR, 675 – 775 nm) and the shortwave infrared range (SWIR, 2314 – 2382 nm), the latter being used to derive methane total columns. The pixel size at nadir was about $7.2 \times 7.2$ km$^2$ before 2019/08/06 and it was upgraded to $5.6 \times 7.2$ km$^2$ after a change in the instrument settings on this date. The swath is about 2600 km on ground, which allows the instrument to provide daily global coverage. The accuracy requirements for XCH$_4$ total columns are 1% bias and 1% random error, as defined in the S5P Calibration and Validation Plan (2017). In this study, the assimilated methane retrievals consist of three level 2 (L2) products, described in the following sections.

**SRON**

The operational TROPOMI product (v2.4), hereafter called "SRON", is developed by the SRON Netherlands Institute for Space Research. The reprocessed product used in this study includes the iterative improvements of the beta research product (Lorente et al., 2021, 2022, 2023). It is retrieved from TROPOMI measurements with the RemoTeC full-physics algorithm, that retrieves both the atmospheric methane mixing ratio and the physical scattering properties of the atmosphere. This algorithm is based on the Philips-Tikhonov regularization scheme, which iteratively optimizes the model parameters so that the model fits the radiance measurements in the SWIR and NIR spectral bands as closely as possible. The state vector is composed of $CH_4$ partial sub-column concentrations, using a priori vertical profiles from the global CTM TM5. The optimized vector is integrated along the vertical dimension to provide as an output the total column-averaged dry-air methane mole fraction XCH$_4$. Cloud-free scenes are identified with the data of the Visible Infrared Imaging Radiometer Suite (VIIRS) instrument from the Suomi-NPP satellite, that flies 3.5 min ahead of S5P. A detailed description of the algorithm is given in the Algorithm Theoretical Baseline Document (Hasekamp et al., 2022).





Recent improvements include better accounting for surface reflectance. We therefore use the reprocessed albedo bias-corrected version and apply the recommended filtering criteria (Apituley et al., 2024) to ensure high-quality data (`qa_value>`0.5). 4,731,034 observations are available on the domain for 2019, including 9.5% over ocean (Table 1). A destriping procedure
(Borsdorff et al., 2024) is applied to new XCH$_4$ data from 2024/09/07 (v2.07), but older orbits have not been reprocessed. The observation errors are only based on the single sounding precision due to measurement noise. We multiply the provided errors by a factor 2 as suggested in the product Readme (Landgraf et al., 2024).

### Blended GOSAT+TROPOMI product

The BLENDED product (Balasus et al., 2023) results from a different approach: a machine learning (ML) model has been trained to predict the differences between GOSAT and TROPOMI observations, based on SRON data features (SRON product v2.4). The GOSAT XCH$_4$ product uses a proxy retrieval which is less sensitive to surface and atmospheric artifacts. It is described in further detail in Section 2.1.2. The global mean bias versus TCCON data (9.2 ppb) is subtracted from all GOSAT observations. The correction is modeled on co-located observations, then applied to the whole TROPOMI-SRON record. This
method aims at taking advantage of both the accuracy of GOSAT measurements and the density of TROPOMI measurements, while mitigating the effect of known limitations of the latter one (biases due to albedo and coarse aerosol particles, striping, etc). We keep only highest quality data (`qa_value= 1`) and filter for coastal scenes as recommended by the authors. This quality filter is theoretically stricter than the one recommended for the SRON product, but in practice it eliminates only 12 observations over the domain in 2019. It can thus be considered that both SRON and BLENDED products share the same
observation sampling (see Table 1). In the BLENDED product, the observation errors, averaging kernels and prior profiles are directly taken from the SRON product (only the XCH$_4$ values differ between the two products), and we multiply the provided errors by a factor 2 (same as for SRON data).

### WFMD

The WFMD scientific product v1.8 (Schneising et al., 2019, 2023), hereafter called "WFMD", is based on the Weighting Functions Modified Differential Optical Absorption Spectroscopy (WFMD-DOAS). This algorithm simultaneously retrieves atmospheric CH$_4$ and CO. It is a least-squares procedure using scaling (or shifting) of previously selected atmospheric vertical profiles to fit the output of a linearised radiative transfer model to the measured spectrum. Quality filtering is based on a random forest (RF) classifier, which is trained to predict high quality measurements using cloud cover from VIIRS. Post-processing
includes systematic bias correction (eg. due to albedo) based on a RF regressor, as well as a destriping filter based on combined wavelet–Fourier filtering.

Data with `xch4_quality_flag`=0 is selected, and 7,627,595 observations are available for 2019, 3.7% of which are over ocean (Table 1). The observation uncertainty is propagated from the noise in the measured spectra, however the unknown noise components related to atmospheric conditions or instrumental features are not accounted for. To avoid underestimating
the uncertainty, the error is rescaled based on a regression of the scatter relative to total columns observations of the ground-based TCCON stations (listed in Appendix C). This rescaling (Equation 1) is included in the product and described in the



Algorithm Theoretical Baseline Document (Schneising, 2023). It leads to different error characterization for SRON/BLENDED and WFMD, as described in Section 2.1.3.

$$\hat{\sigma} = 4/3(\sigma + 5\text{ppb}) \tag{1}$$

| Product | Obs count | Of which ocean |
|---------|-----------|----------------|
| SRON | 4,731,034 | 448,721 (9.5%) |
| BLENDED | 4,731,022 | 448,720 (9.5%) |
| WFMD | 7,627,595 | 283,722 (3.7%) |
| In common | 3,432,335 | 106,775 (3.1%) |
| GOSAT | 12,841 | 790 (6.2%) |

**Table 1.** Number of available observations over the domain in 2019, for the three TROPOMI products SRON, BLENDED and WFMD used in this study, the "common" dataset composed of the observations shared across the three TROPOMI products, and GOSAT.

To separate the effect of varying coverage between products from the effect of the differences in the $XCH_4$ distributions, we extract a dataset of observations shared across all three products, so that they have identical spatio-temporal sampling. This "common" dataset includes observations that match in latitude, longitude and date, within a maximum timelag of $\Delta t = 1$ s (smaller than the duration between consecutive individual observations, $t_o = 1.2$ s) and maximum distance between pixel centers $\Delta x = 0.01°$. It is composed of 3,432,335 observations over 2019. This dataset is used for the analysis of the inter-product differences (Section 3.1) and for OSSEs (Sections 2.4.3, 3.4).

*2.1.2   Other $CH_4$ observations*

In Section 2.1.3, we compare TROPOMI observations with retrievals from the University of Leicester GOSAT Proxy $XCH_4$ product, v9.0 (Parker et al., 2020). The GOSAT satellite was launched in 2009. Iterative improvements of the quality of the retrievals made it a relatively mature product, even though sparse. Methane estimates are based on the $CO_2$ proxy method, which takes advantage of $CO_2$ absorption in the measured 1.65 $\mu$m band, with finer spectral resolution than TROPOMI. This approach significantly reduces the error in the retrieval, since it cancels aerosol and surface artifact contributions that similarly affect $XCO_2$ and $XCH_4$ measurements. Only highest-quality data (quality flag of 0) was considered, yielding 12,841 observations over the domain in 2019. The global mean bias versus TCCON data is removed from the GOSAT observations shown in the following, subtracting 9.2 ppb from all retrievals, to be consistent with the dataset used by Balasus et al. (2023) to derive the BLENDED observations.

To evaluate the results of the inversions, we use independent in-situ measurements of $CH_4$ mixing ratios from flasks and continuous sampling sites. The 19 available surface sites for 2019 are listed in Table A1. The data are compiled from the ICOS atmospheric network (https://www.icos-cp.eu), the World Data Centre for Greenhouse Gases (WDCGG, https://gaw.kishou. go.jp), the NOAA ESRL discrete sampling network (https://www.esrl.noaa.gov/gmd/) and the EBAS data base. The locations of all surface stations are shown in Figure 5. Data from sites with continuous measurements are averaged to hourly values. The





evaluation consists in comparing the emissions derived from the satellite-based inversions to an inversion assimilating only these surface measurements. We also compare the posterior simulated concentrations at the locations of the surface stations to these independent measurements, to see if we can obtain consistent atmospheric values from the satellite and the surface perspectives.

*2.1.3  Comparison of the TROPOMI XCH$_4$ coverage, distribution and errors*

**Coverage** - The inter-comparison of TROPOMI satellite retrievals reveals differences in coverage over Europe in 2019. The density of observations is a critical parameter for ensuring robust constraints on emissions during the inversion. Yet, substantial differences exist between SRON/BLENDED and WFMD, as shown in Table 1 for the total number of data. As BLENDED is a post-processed version of SRON, they have the same sampling. However, WFMD exhibits 63% more observations in
comparison to SRON and BLENDED over Europe in 2019. This higher observation count is attributable to the different filtering approaches: WFMD employs machine learning techniques to identify scenes that are poorly characterized by the retrieval algorithm; quality control criteria for SRON and BLENDED rely on threshold-based filters for scene description variables.

Spatial patterns of differences arise due to differences in quality filtering, as illustrated in Figure 1b. The WFMD dataset
demonstrates a higher density of observations in arid regions (e.g., over Spain and Turkey). Moreover, it includes a few observations in mountainous areas (Alps, Norway), whereas SRON and BLENDED filter out all retrievals in these regions. The ocean coverage is similar, with a decrease in the density for latitudes above 48°. WFMD is the only product which includes observations in the North Sea, a region of interest due to detectable emissions from oil and gas platforms (Riddick et al., 2019).

The covered period includes the change in pixel size, starting on the 6th of August 2019, due to an increase of the spatial
resolution (7.2 km to 5.6 km) in the along track direction. The temporal coverage shows consistent variations for the three products (Figure 1a): observation density is high between June and October and decreases during winter. An anomalously low density is noticeable for all products in May 2019. It is likely to be associated with an increase of the Fractional Cloud Cover (CFC) over Europe during this month, as indicated by the CLARA-A3 record (Karlsson et al., 2023). Seasonal spatial variations are consistent among products, with a marked absence of observations at high latitudes ($\gtrsim 60°$) during winter months
(Figure S1).

**Comparison of TROPOMI observations** - The XCH$_4$ distributions exhibit approximately Gaussian shapes, with averages of respectively 1850, 1843 and 1856 ppb for SRON, BLENDED and WFMD (Figure 3a). BLENDED lower concentrations align well with the distribution of GOSAT observations for the same year (mean XCH$_4$ of 1843 ppb). It is interesting to note that without the removal of the mean bias of GOSAT with respect to TCCON observations (-9.2 ppb), the distribution of
GOSAT observations would have been closer to those of SRON and WFMD. Part of the differences between the distributions can be explained by the differences in spatial coverage between SRON/BLENDED and WFMD. However, when considering the common observations across the three datasets, the distributions are slightly closer but still differ: the average XCH$_4$ are 1851, 1844 and 1856 ppb respectively for SRON, BLENDED and WFMD.





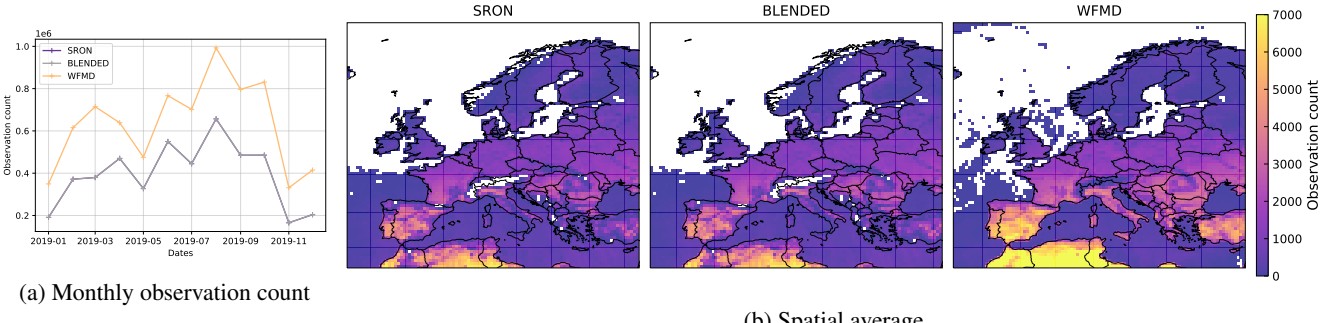

(a) Monthly observation count

(b) Spatial average

**Figure 1.** Number of observations per month (panel **a**) and per 0.5°latitude/longitude bin in 2019 (panel **b**) for the three TROPOMI products SRON, BLENDED and WFMD used in this study.

Overall, the spatial patterns of the distributions of CH$_4$ concentrations are consistent, yet some local discrepancies exist,
e.g., in Scandinavia (Figure 2b). BLENDED show lower XCH$_4$ over the whole domain, but also consistent spatial gradients
with SRON and WFMD (Figures 2b, S2). The temporal variations of XCH$_4$ throughout the year show consistent patterns
across the products and align rather well with GOSAT (Figure 2a). Average concentrations decrease slightly from January to
May, reaching a minimum in April/May, before rising during late summer and autumn, peaking in November. This pattern
is consistent with previous analyses of the CH$_4$ seasonality: the late summer increase is primarily attributed to wetlands
emissions, while the higher concentrations in winter are associated with reduced loss from the hydroxyl (OH) radical (Vara-
Vela et al., 2023; East et al., 2024). BLENDED and WFMD have similar variations, but an offset of approximately 10 ppb.
SRON observations tend to be closer to GOSAT and BLENDED in winter, but closer to WFMD in summer. Detailed study of
the differences of XCH$_4$ relative to TCCON is available in the supplementary material (Appendix C).

To better understand the drivers of the differences in XCH$_4$ distributions, we adopt the method outlined by Balasus et al.
(2023). A machine learning (ML) model is trained to predict the difference in XCH$_4$ based on the retrieval variables. The
contribution of individual features to the prediction is then assessed to estimate the importance of each variable. The results
and analysis of the biases are presented in Section 3.1.

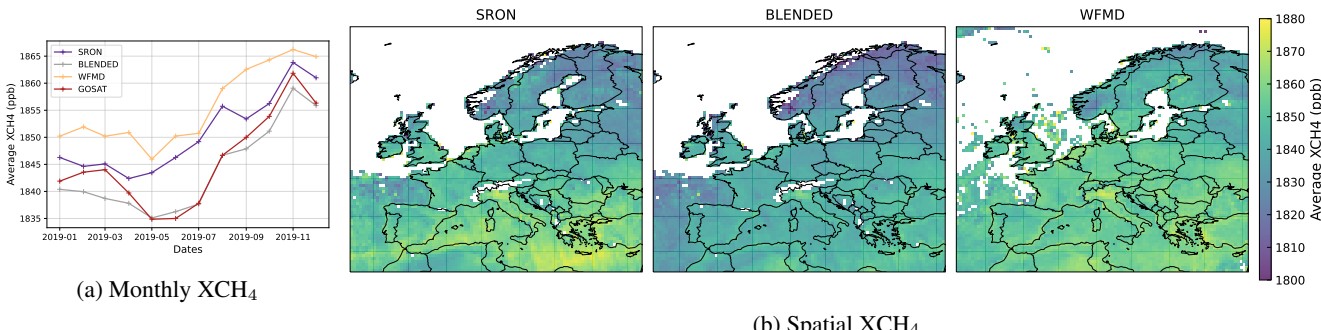

(a) Monthly XCH$_4$

(b) Spatial XCH$_4$

**Figure 2.** Average XCH$_4$ observation per month (Panel **a**) and per 0.5°latitude/longitude pixel in 2019 (Panel **b**) for the three TROPOMI products.



**Observation error** - The error attributed to individual observations differs between the SRON/BLENDED and WFMD products (Figure 3b). As explained in Section 2.1.1, the error reported for SRON (thus also for BLENDED) represents the standard deviation of the retrieval noise (Lorente et al., 2021) and only includes the effect of noise in the measured radiances. The WFMD errors are rescaled with the linear transformation of Equation 1, to better estimate the uncertainty and include the contribution of the pseudo-noise component determined by atmospheric or instrumental features (Schneising, 2023).

Despite the suggested rescaling of the error by a factor of 2 (Apituley et al., 2024), the errors for SRON and BLENDED (3.9 ppb in average) are an order of magnitude smaller than those for WFMD (12.2 ppb). When the linear transformation of Equation 1 is applied to SRON/BLENDED errors, the resulting error distribution aligns more closely with WFMD: the average scaled error is 11.9 ppb for SRON and BLENDED (Figure 3c). The difference of error definition in the retrieval procedure is critical, as the observation errors provided by these products do not combine the same error sources and therefore have different orders of magnitude. The impact of this difference on the results of inversions is further evaluated in Section 3.4.

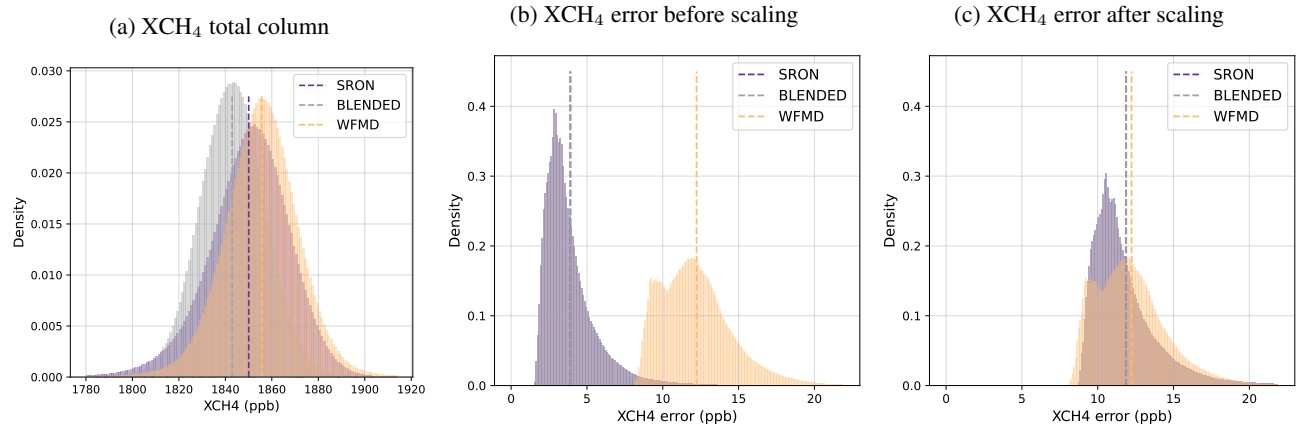

**Figure 3.** Distributions of observations and observation errors (both in ppb) for the three TROPOMI products. For the error, panel **(b)** shows the raw errors, while panel **(c)** shows the same histogram after error scaling for the SRON and BLENDED products.

**Vertical parameters** - As illustrated in Figure 4 (only the common observations are considered here), the averaging kernels (AKs) and prior profiles exhibit similar shapes, typical of SWIR retrievals. The relative difference between SRON/BLENDED and WFMD vertical profiles is generally limited to 0-5%, though the differences can lead to large differences in the computed simulated equivalents of $XCH_4$. The WFMD AKs seem to be less sensitive to the layers close to the surface in comparison to SRON/BLENDED, but more sensitive to the stratosphere (pressures inferior to 200 hPa). Differences up to approximately 200 ppb occur between the prior profiles of SRON/BLENDED and WFMD, in particular for the layers close to the surface: the WFMD prior profiles are scaled to have uniform surface values of 1850 ppb, while SRON/BLENDED prior mixing ratios at the surface are between 1900 and 2000 ppb. The scaling of prior profiles for WFMD also explains the lower variability of prior mixing ratios in comparison to SRON/BLENDED, as evidenced by the narrower horizontal deviations in the right panel of Figure 4. These profiles are consistent with the ones presented in Lindqvist et al. (2024), which also highlight a systemically



higher prior profile for SRON in comparison to the one of WFMD. The differences in the profile leads to differences in the
simulated XCH$_4$ distributions, as shown in Section 3.2.

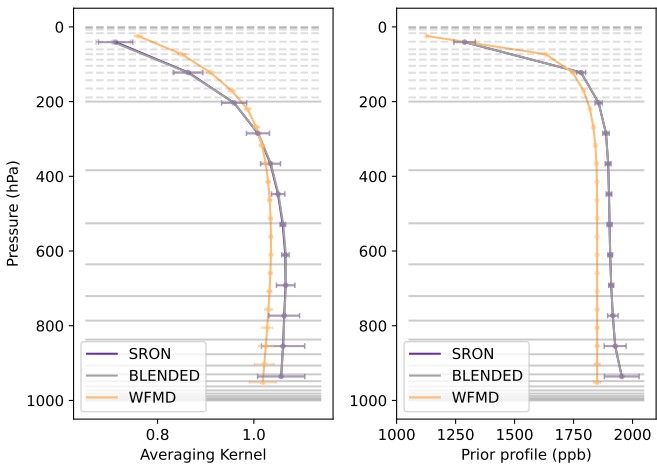

**Figure 4.** Averaging kernels (left, unitless) and prior profiles (right, in ppb) averaged between the common observations of the TROPOMI products (dataset described in Section 2.1.1). SRON and BLENDED have exactly the same profile, since the only difference in the co-located dataset is the XCH$_4$ value. Horizontal lines are CHIMERE pressure levels (plain lines) and CAMS pressure levels in the stratosphere (dashed lines, for pressures lower than 200 hPa), for a surface pressure of 1000 hPa.

## 2.2 Prior estimates of the CH$_4$ emissions

Prior methane emissions are compiled from several BU inventories. Anthropogenic emissions are from EDGARv8.0 (Crippa et al., 2023). The biomass burning fluxes are taken from GFEDv-4.1s (Randerson et al., 2018). Natural fluxes consist of an ensemble of datasets provided by the Global Methane Budget protocol for inversions (GCP-CH$_4$, Saunois et al., 2020). Fluxes
from wetlands (peatlands, inundated and mineral soils) are from the JSBACH-HIMMELI model (Raivonen et al., 2017). The geological emissions are a climatology based on Etiope et al. (2019) and scaled down to a global total of 15 TgCH$_4$ yr$^{-1}$ in accordance with the maximum suggested by Petrenko et al. (2017). The emissions due to termites are a climatology based on the estimate of S. Castaldi, from GCP-CH$_4$ (Saunois et al., 2020). Finally, the ocean fluxes are a climatology based on Weber et al. (2019). All these datasets have been interpolated at the 0.5°×0.5° horizontal resolution of the CTM grid. The map of total
emissions in 2019 is shown in Figure 5. Anthropogenic emissions contribute to about 72% of the total budget over the domain, which is about 43.6 TgCH$_4$.

## 2.3 Configuration of the CHIMERE CTM for the simulation of CH$_4$ concentrations

The CTM used for inversion is the regional chemistry-transport model CHIMERE (Menut et al., 2013; Mailler et al., 2017) and its adjoint code (Fortems-Cheiney et al., 2021). The targeted domain covers all European Union (EU) countries: it spans
from 15°W to 35°E and 32°N to 74°N. This domain has already been used for the intercomparison of inversions EUROCOM



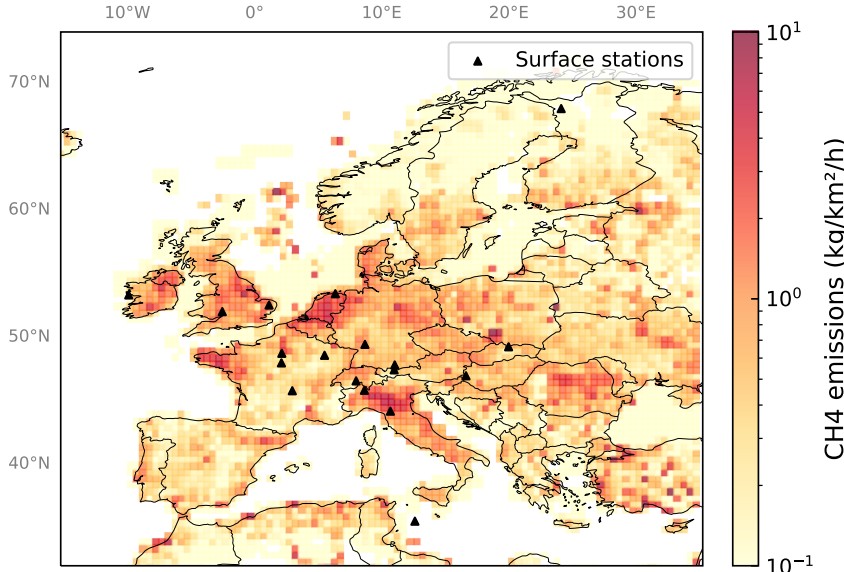

**Figure 5.** Prior CH$_4$ emissions (kg.m$^{-2}$.h$^{-1}$) in Europe in 2019, in log-scale. Black triangles are the locations of the surface stations listed in Table A1.

(Monteil et al., 2020). The grid used for discretizing emissions, meteorological data and other inputs and for running the CTM covers this domain at a 0.5° longitude × 0.5° latitude resolution. Vertically, the domain where transport is simulated by CHIMERE extends from the surface to 200 hPa (with 17 sigma-pressure levels). Above this pressure (approximately the tropopause) and up to 0.1 hPa, CH$_4$ concentration fields are taken from the CAMS reanalysis simulations (based on the
Integrated Forecasting System (IFS) system of the European Centre for Medium-Range Weather Forecasts (ECMWF)) (Agustí-Panareda et al., 2023). Lateral boundary and initial conditions of methane mixing ratios are taken from the same CAMS product.

CHIMERE requires a set of meteorological variables, taken from the ECMWF IFS operational forecast (every three hours) retrieved at 0.25°×0.25° and interpolated onto the model's grid. The size of the domain makes it possible to neglect the chemistry of CH$_4$ because its oxidation by hydroxyl radicals leads to a lifetime from 8 to 10 years (Saunois et al., 2020),
whereas the ventilation time of the domain is generally less than two weeks.

To ensure relevant comparisons between simulated concentrations and TROPOMI observations, the equivalents of the observations are post-processed from the output 4D field of CHIMERE, within the CIF. The AKs are applied to the model profiles to compute the simulated XCH$_4$ $\hat{y}$ using the formula provided in the user's manual of the TROPOMI products (Apituley et al., 2024; Schneising, 2022):

$$\hat{y} = \sum_l h^l \left( y_a^l + a_l(y_s^l - y_a^l) \right) \qquad (2)$$





where $l$ is the index of the vertical layer of TROPOMI, $\boldsymbol{h}$ is the vector of pressure weights ($h^i = \Delta p^i / \sum_l \Delta p^l$) with $\Delta p^l$ the difference of pressure between the bottom and the top of layer $l$. $\boldsymbol{a}$ is the column averaging kernel, $\boldsymbol{y}_a$ and $\boldsymbol{y}_s$ respectively the prior and simulated dry air mole fraction vertical profiles, the latter being interpolated on the TROPOMI levels. This vertical interpolation from CHIMERE to TROPOMI levels is linear and pressure weighted.

### 2.4 Variational inversions in the CIF-CHIMERE inversion system

#### 2.4.1 Principle of Bayesian variational inversion

In the following, we use notations according to the convention defined by Ide et al. (1997) and Rayner et al. (2019). The inversion adjusts the control vector $\boldsymbol{x}$, to improve the fit between observed satellite data and their simulated equivalents. The set of control parameters is composed of the $CH_4$ emission maps whose estimate is the primary target of the inversion, and the 4D background field used to impose the initial, lateral and top boundary conditions as well as the concentrations above 200 hPa. A priori information about the control variables is given by the vector $\boldsymbol{x}^b$.

The 4D variational scheme relies on a classical Bayesian framework and optimizes the control vector so that it minimizes the cost function $J$. This function is the sum of two contributions, induced respectively by the difference between prior and posterior estimates and by the difference between simulated and observed concentrations:

$$J(\boldsymbol{x}) = \frac{1}{2}(\boldsymbol{x} - \boldsymbol{x}^b)^T \mathbf{B}^{-1}(\boldsymbol{x} - \boldsymbol{x}^b) + \frac{1}{2}(\mathcal{H}(\boldsymbol{x}) - \boldsymbol{y}^0)^T \mathbf{R}^{-1}(\mathcal{H}(\boldsymbol{x}) - \boldsymbol{y}^0) \tag{3}$$

$\boldsymbol{x}^b$ and $\boldsymbol{y}^0$ are respectively the vector of prior information and the vector of observations. $\mathcal{H}$ is the observation operator that projects the control vector $\boldsymbol{x}$ onto the observation space, so that $\mathcal{H}(\boldsymbol{x})$ is the simulated equivalent of observations based on the CTM CHIMERE. The matrices $\mathbf{B}$ and $\mathbf{R}$ define respectively the error covariances of the control vector and observation errors. The latest combines errors in both the observation data (measurement or processing errors) and the observation operator (model error, representativity of the gridded model compared to point measurements, aggregation errors).

#### 2.4.2 Variational inversion procedure

The CIF is a modular inverse modeling platform developed as a python library (Berchet et al., 2021), designed in the framework of European and international projects. It can drive various data assimilation schemes (analytical inversions, Ensemble Kalman filtering and 4D variational inversions) and it can be coupled to various CTMs. In this study, we run 4D-Var inversions. The cost function is minimized using the quasi-newtonian M1QN3 algorithm (Gilbert and Lemaréchal, 1989). The inversion is stopped when the gradient norm reduction exceeds 95% compared to its initial value, usually corresponding to between 20 and 30 iterations.

The control vector $\boldsymbol{x}$ contains $CH_4$ emissions, at a $0.5°\times0.5°$ horizontal resolution and a weekly temporal resolution (the spatio-temporal grid is similar to those of Petrescu et al. (2024) and Szénási et al. (2021)). It also contains the 4D $CH_4$ background field used to impose the initial and boundary conditions and the concentrations in the stratosphere (above 200 hPa) at





the native pixel resolution of the corresponding CAMS product ($3° \times 2°$) and 2-day temporal resolution. Indeed, the assimilated data is used by the inversion to retrieve information on emission fluxes but also on the background.

The error covariance matrix $\mathbf{B}$ is built by block as follows, the errors on respectively the emissions and the background being considered independent. For the $CH_4$ emission components of the control vector ($0.5° \times 0.5°$ spatial resolution, weekly
temporal resolution), the standard deviation is set at 100%. Spatial and temporal covariances between these components are taken into account as an e-folding decrease with correlation lengths of 150 km on land and 200 km on sea, and 2 weeks through time. The second block of $\mathbf{B}$ is set for $CH_4$ background concentrations ($3° \times 2°$ spatial resolution, 2-day temporal resolution): we assume a 2% standard deviation for diagonal elements. The non-diagonal elements account for spatial and temporal covariances, using a similar e-folding decrease with a spatial correlation length of 200 km and a temporal correlation
length of 14 days.

In the matrix $\mathbf{R}$, there is no correlation of the errors from one observation to another, i.e. $\mathbf{R}$ is diagonal. The diagonal elements are the errors associated with the individual retrievals, as described in Section 2.1.3.

For the inversions, additional filters are applied to TROPOMI observations (Section 2.1.1) to avoid large differences between observed and simulated $XCH_4$ caused by a large model error. The CTM cannot capture subpixel pressure variations, thus
surface roughness is a source of poor model representation: to mitigate the impact of subpixel topographic variability, we remove observations with surface pressure deviating by more than $3\sigma$ from the mean pressure of the data aggregated at the scale of the CTM pixel. This filter eliminates 1.4% of the observations for SRON and BLENDED, 1.0% for WFMD. Data for which the difference between the observation and the simulation is more than 100 ppb are also filtered out. It impacts very few observations: this second filter removes respectively 420 observations for SRON/BLENDED and 934 for WFMD, i.e 0.009%
and 0.012% of the data. The spatial distributions of the observations removed by these two filters are shown in Figure D1, they are similar across the three products. It confirms that the discrepancies are due to the CTM and not the observations.

In this study, we perform four inversions assimilating real observations. The first three inversions assimilate separately each one of the TROPOMI products. An additional inversion assimilating surface observations (stations listed in Table A1) is performed for evaluation. These four inversions are listed in Table 2.

| ID | Observation type | Observation product |
|---|---|---|
| Inv-SRON | Satellite | SRON |
| Inv-BLD | Satellite | BLENDED |
| Inv-WFMD | Satellite | WFMD |
| Inv-Surface | Surface | Hourly data/flasks from the stations listed in Table A1 |

**Table 2.** Synthesis of the inversions performed in this study. Results are presented in Section 3.3.

### 2.4.3  *Observing System Simulation Experiments*

We also explore the capability of the inversion set-up to derive robust emission estimates based on the assimilation of TROPOMI
products in a set of Observing System Simulation Experiments (OSSEs). The OSSE approach (Lahoz and Schneider, 2014)





makes it possible to evaluate the sensitivity of the inversions to the observation coverage, the observation errors and the differences between products, comparing the results of multiple OSSE scenarios. The structure of an OSSE is described in Figure 340 6. The general principle consists of two main parts (Brasseur and Jacob, 2017): first, the sampling of "true" synthetic observations, given a true state defined by the prior emissions and background concentrations. Then, a Monte-Carlo ensemble of inversions with perturbed priors is performed, to evaluate the capability of the observing system to recover the true state of the emission and background estimates.

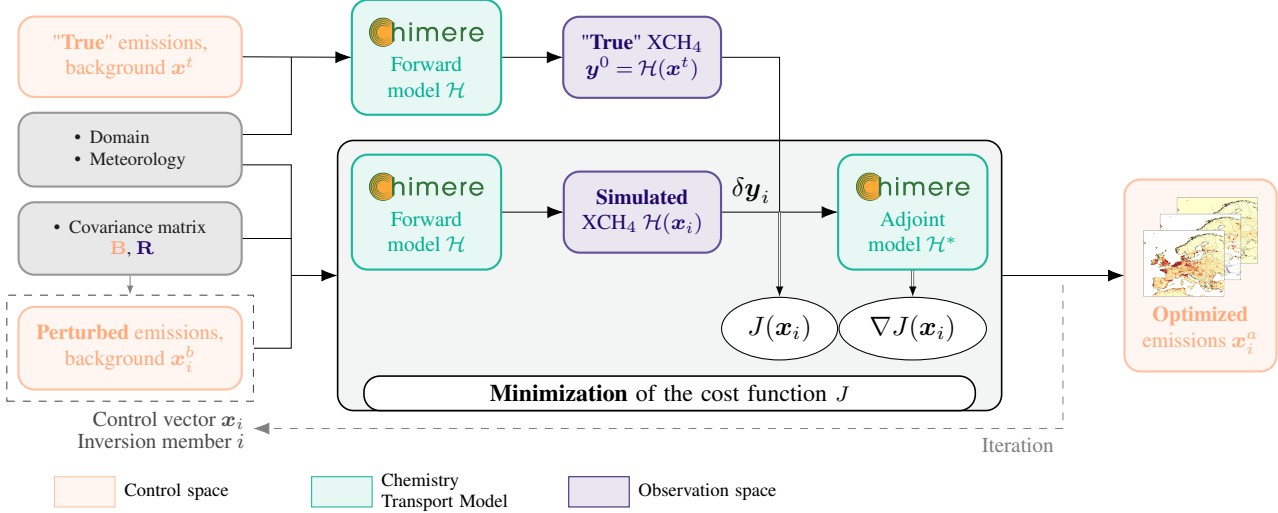

**Figure 6.** OSSE structure within the CIF. "True" pseudo-observations $\boldsymbol{y}^0$ are synthesized from the prior emissions $\boldsymbol{x}^t$ and then assimilated in an ensemble of inversions using perturbed prior fluxes $\boldsymbol{x}_i^b$. See http://community-inversion.eu.

In more details, a forward run of the CHIMERE CTM produces a "true" 4D concentration field, based on the prior emissions and the prior background concentrations (called $\boldsymbol{x}^t$). Pseudo-observations $\boldsymbol{y}^0$ are sampled from this field following the specifications (date, location, averaging kernels...) of the observation dataset.

The pseudo-observations are assimilated in an ensemble of $n = 4$ inversions using randomly perturbed priors: for each inversion member $i$, the priors are initially perturbed (called $\boldsymbol{x}_i^b$) according to the error statistics provided in $\mathbf{B}$ (Section 2.4.2):

$$\forall i \in [1, n], \boldsymbol{x}_i^b = \boldsymbol{x}^t + \boldsymbol{\eta}_i, \quad \boldsymbol{\eta}_i \sim \mathcal{N}(0, \mathbf{B}) \tag{4}$$

The metrics used to assess the capability of the system to improve the emission estimates (i.e. to make them closer to the truth) is the relative increment $r$. It compares the distance to the true initial fluxes $\boldsymbol{x}^t$ of the perturbed prior $\boldsymbol{x}_i^b$ and the posterior fluxes $\boldsymbol{x}_i^a$, averaged over all the members of the ensemble:

$$\boldsymbol{r} = \frac{1}{n} \sum_i \boldsymbol{r}_i = \frac{1}{n} \sum_i \left( \left| \frac{\boldsymbol{x}_i^a - \boldsymbol{x}^t}{\boldsymbol{x}_i^b - \boldsymbol{x}^t} \right| - 1 \right) \tag{5}$$



The closer $r$ is to $-100\%$, the more the inversion improves the fluxes. Negative values of $r$ correspond to posterior emissions that are closer to the truth than the prior (thus "better" emissions) while positive values correspond to posterior emissions further from the truth.

We define 6 OSSE scenarios, listed in Table 3, using different pseudo-observation datasets. To ensure a robust comparison of the scenarios, the seed of the random perturbations is similar in all six. It is important to note that since SRON and BLENDED share the same sampling and parameters (errors, vertical profiles), the pseudo-observation datasets are identical. In the following, *"SB"* scenarios stand for the two products. The 6 scenarios are designed to evaluate the sensitivity of the inversions to multiple parameters:

- 3 OSSEs are performed as reference scenarios using the pseudo-observations of the TROPOMI products: *"Ref-SB"* for SRON and BLENDED, *"Ref-WFMD"* for WFMD; a third reference OSSE (*"Ref-Surf"*) is performed using the pseudo-observations of the surface stations.

- To evaluate the sensitivity to the observation density, 2 OSSEs are carried out assimilating only the pseudo-observations corresponding to the common observations (as described in Section 2.1.1). The pseudo-observation datasets differ only by the observation errors and vertical profiles (prior profiles and averaging kernels). These scenarios are referred to as *"Common-SB"* and *"Common-WFMD"*.

- To evaluate the sensitivity to observation errors, one scenario is defined using the SRON/BLENDED pseudo-observations dataset, rescaling the errors with Equation 1. It is called *"Err-SB"*.

To evaluate the impact of the differences between products, we carry out two more inversions assimilating pseudo-observations. The method differs from previous OSSEs: in this case, synthetic observations are sampled and biased with the difference between products:

- The difference WFMD-SRON (computed over the common observations) is averaged over $0.5°\times0.5°$ pixels and 1-hour-long periods, and added to the corresponding SRON pseudo-observations.

- The same correction is applied to WFMD pseudo-observations with respect to SRON-WFMD difference.

- Each "corrected" data set is assimilated (priors are not perturbed), the inversions are called *"Diff-SRON"* and *"Diff-WFMD"*. This method makes it possible to evaluate the impact of the inter-product difference on the posterior increments.





| ID | Product | Obs subset | Error correction | Difference correction |
|---|---|---|---|---|
| Ref-SB | SRON/BLENDED | All obs | No | No |
| Ref-WFMD | WFMD | All obs | No | No |
| Ref-Surf | Surface | All obs | No | No |
| Common-SB | SRON/BLENDED | Common obs | No | No |
| Common-WFMD | WFMD | Common obs | No | No |
| Err-SB | SRON/BLENDED | Common obs | Yes | No |
| Diff-SRON | SRON/BLENDED | Common obs | No | Yes (w.r.t. WFMD) |
| Diff-WFMD | WFMD | Common obs | No | Yes (w.r.t. SRON) |

**Table 3.** Synthesis of all the OSSEs performed in this study. "Common obs" refers to the restriction to observations common to the 3 products, as described in Section 2.1.3. The "error correction" corresponds to the application of Equation 1 to the errors of SRON/BLENDED.




## 3 Results

### 3.1 Drivers of the differences in the XCH$_4$ observed distributions

To better understand the inter-product differences related to atmospheric, instrumental and algorithmic variables, we adopt the method of Balasus et al. (2023): we train a ML model to predict the $\Delta$XCH$_4$ differences between observed values, for each pairwise combination of products, using retrieval parameters as input features. Ten features are included: surface altitude and roughness, SWIR surface albedo, fluorescence, the aerosol size parameter, the SWIR aerosol optical thickness (AOT), the a priori XCH$_4$ total column and the across-track pixel index, which are retrieved from the SRON product, and the observation errors of the two products for which $\Delta$XCH$_4$ is predicted. The across-track pixel index provides information about the relative position of the pixel in the swath of the satellite, therefore it is related to the so-called "striping" effect — a systematic artifact observed in TROPOMI CO, H$_2$O/HDO, and XCH$_4$ products (Borsdorff et al., 2019). This effect consists in differences of XCH$_4$ measurements across consecutive parallel strips. The 3,432,335 observations shared across the three products (dataset described in Section 2.1.1) are randomly split into a training dataset (90% of the observations) and a validation dataset (10% of the observations). We search for the best-performing model between three ML algorithms that rely on decision trees: Random Forest, XGBoost and Light Gradient-Boosting Machine (LightGBM). Random Forest builds an ensemble of decision trees by combining bootstrap sampling and feature randomization, and produces predictions through averaging across the forest (Breiman, 2001). XGBoost and LightGBM apply gradient-boosted decision trees, a method where sequentially trained decision trees minimize a loss function by correcting the residual errors of the previous trees, thereby iteratively enhancing accuracy. XGBoost employs advanced regularization (L1 and L2 penalties) and parallelized tree construction (Chen and Guestrin, 2016). In contrast, LightGBM adopts a histogram-based learning approach and exclusive feature bundling (Ke et al., 2017).

In all cases, LightGBM outperforms the other models, while Random Forest systematically shows the lowest performances. Table 4 presents the validation results for the LightGBM model, which was ultimately chosen for further analyses. The predictive performances are consistent with the results of Balasus et al. (2023) for the prediction of the XCH$_4$ differences between TROPOMI and GOSAT, in terms of RMSE (Root Mean Square Error) and correlation (RMSE= 12.4 ppb, $R^2 = 0.53$).

Beyond its predictive performances, the trained model provides insights into the most important features — that is, the features that contribute most to the prediction of $\Delta$XCH$_4$. We conduct a study using SHapley Additive exPlanations (SHAP) on the predictions of the model, following Balasus et al. (2023). This approach partitions an individual prediction into contributions attributed to each feature, quantified as SHAP values (in ppb). The sum of all SHAP values for a given prediction corresponds to the difference between that prediction and the mean prediction across the entire dataset. The input features can then be ranked using the average SHAP values, for each product pair comparison (Figure 7). Our findings align closely with those of Balasus et al. (2023): the most impacting features on the differences between satellite products are the across-track pixel index, aerosols and SWIR albedo. The observation errors also seem to impact the prediction. Features related to aerosols are the SWIR aerosol optical thickness (AOT) and the aerosol size parameter, which are strongly anti-correlated ($R^2 = -0.82$). Since the SHAP analysis does not fully resolve correlations between variables, we aggregated the contributions of these two features into a single combined SHAP value, hereafter called "aerosols", to preserve interpretability.





| Dataset 1 | Dataset 2 | RMSE (ppb) | Correlation ($R^2$) |
|-----------|-----------|------------|---------------------|
| SRON | WFMD | 8.1 | 0.58 |
| SRON | BLENDED | 4.4 | 0.86 |
| WFMD | BLENDED | 8.5 | 0.44 |

**Table 4.** Validation results of predicted $\Delta XCH_4$ for each combination of products, with the best-performing model (Light GBM).

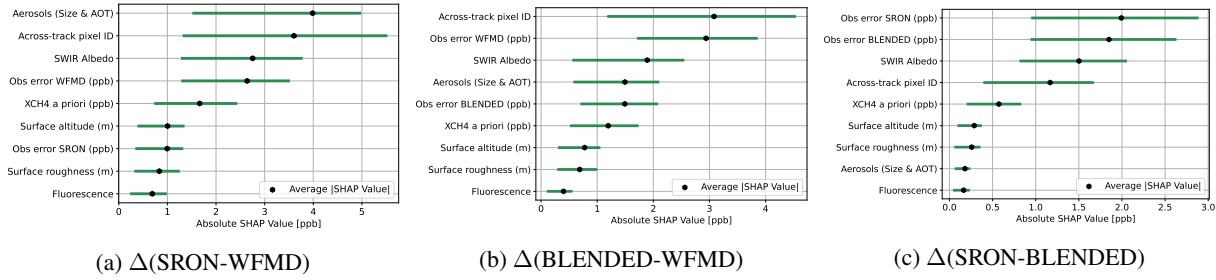

(a) $\Delta$(SRON-WFMD)    (b) $\Delta$(BLENDED-WFMD)    (c) $\Delta$(SRON-BLENDED)

**Figure 7.** Average SHAP values and the interquartile range (green bar), for the 9 most impacting features.

Digging deeper into the impact of these 3 parameters on the products, Figure 8 illustrates the variations in measured $XCH_4$ as functions of SWIR albedo, aerosol size parameter, and across-track pixel index. The albedo and aerosol parameters are retrieved with the SRON algorithm: it is important to mention that the accuracy of these retrievals is not clearly known.

– **Albedo** - $XCH_4$ are influenced by surface reflectance, particularly in scenes with low SWIR albedo (e.g., snow, water bodies) or high SWIR albedo (sandy areas). In all the products, the albedo dependence is corrected based on a polynomial fit to the surface reflectance spectrum in the 2.3 $\mu$m spectral range. The polynomial degree was increased from 2 to 3 in recent updates of the products (Lorente et al., 2023; Schneising et al., 2023). Despite these adjustments, the products show different variations with SWIR albedo, as illustrated in Figure 8a. BLENDED and WFMD have a consistent low dependency on the albedo in the range of values between 0.05 and 0.5, where the density of observations is high. However, there is an offset of approximately 10 ppb, which decreases at high albedo values. SRON $XCH_4$ observations show more variations: they are close to those of WFMD at low albedo values, but lower than the two other products for high albedos.

– **Aerosols** - The effect of aerosols on methane retrievals is well-known: scattering by aerosol particles modifies the light path and induces errors that can compromise the accuracy of the retrieval (Butz et al., 2012). It depends on aerosol amount, type and size distribution. The latter is characterized by the aerosol size parameter $\alpha$, which is the negative exponent of the power law of the aerosol size distribution $n(r) \propto r^{-\alpha}$, with $r$ the particle radius: the higher $\alpha$ is, the more $n(r)$ is shifted towards small particle sizes. It is simultaneously inferred and corrected during the $XCH_4$ retrieval process of SRON (Hasekamp et al., 2022), yet it still strongly impacts $XCH_4$ observations. TROPOMI SRON data is biased low relative to GOSAT for low values of $\alpha$, indicating large particles (Balasus et al., 2023), as confirmed by Figure 8b. The correction applied in the BLENDED product makes its sensitivity to $\alpha$ closer to the one of WFMD,





in comparison to SRON. Still, the three products show different behaviors and seem to be consistent only for $\alpha > 4.4$, which corresponds to a small part of the observations.

– **Striping patterns** - Without dedicated destriping, stripes in the flight direction are visible in the TROPOMI XCH$_4$ data, likely due to variations in the offsets and gains of detector pixels. While the across-track pixel index is included in the calibration process of the WFMD retrieval, residual vertical stripes persist in the data (Schneising et al., 2023). The latest WFMD product (v1.8) mitigates this effect using a wavelet–Fourier decomposition and filtering (Schneising et al., 2023). Indeed, Figure 8c) depicts the higher sensitivity of the SRON product to this effect. The correction with respect

to GOSAT applied in the BLENDED product yields slightly lower sensitivity regarding the striping effect (Figure 8c). A recent method based on a double moving median smoothing along both the flight and cross-track directions resulted in promising enhancements in striping correction for the operational SRON product. It is implemented into newer versions of the product (Borsdorff et al., 2024), but older orbits have not been reprocessed. Future uses of the SRON destriped data should improve the consistency of XCH$_4$ observations with the other products.

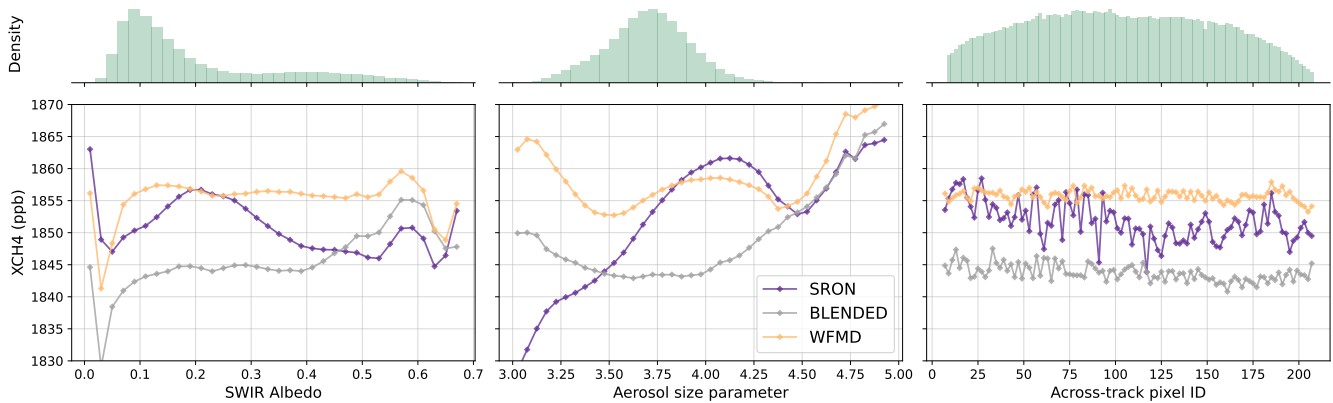

**Figure 8.** Observed XCH$_4$ total column vs. SWIR albedo, aerosol size parameter and across-track index. These three parameters are retrieved with SRON algorithm. The histogram depicts the density of common observations with respect to the corresponding variable.

## 3.2    Comparison between observed and simulated CH$_4$ total columns

The differences in the simulated XCH$_4$ between the three TROPOMI products are linked to the differences in observation coverage and in the vertical profiles that are necessary to compute their simulated equivalents, as described in Section 2.1.1. The mean biases (MBs) of the difference $\Delta$XCH$_4^{os}$ = XCH$_4^{obs}$ − XCH$_4^{sim}$ with the prior inputs are approximately 6.7, -0.4 and 4.4 ppb and the RMSEs 18.2, 13.6 and 15.4 ppb respectively for SRON, BLENDED and WFMD (top row of Figure

9). BLENDED demonstrates the best agreement between simulations and observations, with both a smaller bias and a lower RMSE compared to SRON and WFMD.



When the analysis is restricted to the subset of matching observations across the three datasets, the RMSE decreases for all products, to respective values of 17.4, 13.1 and 13.0 ppb. This improvement indicates that the application of combined filters, which more rigorously select high-quality data, reduces the gap between observed and simulated concentrations.

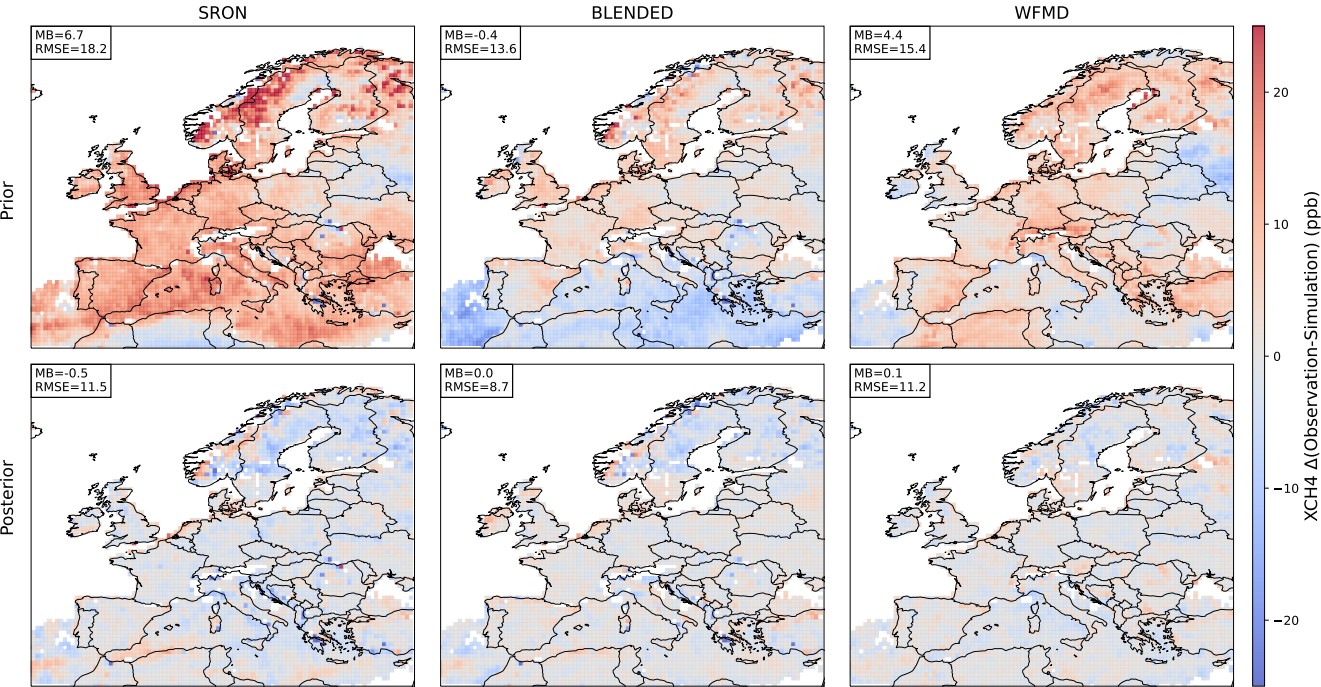

**Figure 9.** Annual average difference between TROPOMI observations and their CHIMERE simulated equivalents, using prior emissions (first row) and posterior emissions (second row). MB is the mean bias and RMSE the root mean square error, in ppb. Only the cells with at least 50 observations in 2019 are shown.

The spatial distributions of the differences between observed and simulated concentrations $\Delta XCH_4^{os}$ (first row of Figure 9) reveal common patterns but also notable discrepancies across the three TROPOMI products. For all products, simulated $XCH_4$ are overall lower than observations over land. SRON has the highest difference between observed and simulated $XCH_4$ over land, especially in Western Europe. Over the sea, the simulations behave very differently across products: SRON shows a positive difference, while it is negative for BLENDED and close to 0 for WFMD. The difference between SRON and BLENDED

directly comes from the differences in the observations, and is consistent with the systematic downward correction over the ocean depicted in Balasus et al. (2023).

Seasonal variations in the $\Delta XCH_4^{os}$ are generally consistent across the three products (Figure D3). $\Delta XCH_4^{os}$ are high during the first half of 2019 and decrease in the second half of the year; the RMSE decreases by more than 20% from winter (i.e. January to March) to summer (i.e July to September) for all products, though this overall reduction masks spatial heterogeneities.

Scandinavia stands out as a region with marked discrepancies between observations and simulations: for SRON, $\Delta XCH_4^{os}$ reaches high positive values from April to June and is negative between October and December (Figure D3). The seasonal





variations are consistent across the three products in this region, though with lower amplitude for BLENDED and WFMD. The differences in behaviour among products in regions like North Africa and Scandinavia underscore the influence of surface albedo and aerosols on observations. This effect predominantly affects observed concentrations, as simulated equivalents
exhibit limited dependence on the parameters identified in Section 3.1 (not shown).

Overall, simulated concentrations are underestimated compared to observations over land across all TROPOMI products, with BLENDED providing the closest match. The inversion process correctly fits the simulated $XCH_4$ to observations: the mean biases are decreased from the prior to the posterior simulation by approximately 93%, 95% and 99% for SRON, BLENDED and WFMD respectively (bottom row of Figure 9). Similarly, the RMSE decreases by 37%, 36%, and 28% for these products.

## 3.3   TROPOMI-derived $CH_4$ emissions : spatial distribution, seasonal cycles and annual budgets

Inversions are performed using the method detailed in Section 2.4.2. The three inversions assimilating each TROPOMI product, as well as the additional inversion using surface data for evaluation, are listed in Table 2. Figure 10 shows the corrections to the prior $CH_4$ emissions from these inversions, referred to as increments, both at the grid-cell and national resolutions.

At the grid-cell resolution, the increments differ among the inversions (Figure 10a). These differences result in large differ-
ences in the total $CH_4$ emission budgets for 2019, for the regions and the corresponding countries listed in Table B1. While the prior total emissions of 25.2 $TgCH_4$/yr are increased by 2% for the SRON (25.7 Tg/yr) inversion, they are reduced by respectively 1%, 9% and 33% for the BLENDED (25.0 Tg/yr), surface-based (23.0 Tg/yr) and WFMD (16.9 Tg/yr) inversions. The detailed comparison of prior and posterior emissions per country is shown in Table S1. While some regions exhibit consistent corrections across all three TROPOMI inversions (e.g. in Northern Africa, Italy or Romania), the magnitude of increments
is generally larger in the WFMD-based inversion in comparison to SRON and BLENDED, consistently with the OSSEs of Section 3.4. Aggregation at the national scale mitigates some of these inconsistencies but reveals high differences in countries such as the Netherlands, Germany, France, and Poland (Figure 10a). SRON and BLENDED provide similar corrections to the prior $CH_4$ emissions in most countries, except Poland and Turkey, but they differ from WFMD particularly in Western and Central Europe.

It is important to note that the surface-based inversion is influenced by the spatial distribution of the stations: regions with dense station coverage, such as Western Europe (8 stations), are well constrained, whereas regions with sparse coverage, like the Baltic states, Spain and Portugal show little correction of the prior fluxes. Scandinavia is poorly constrained in all satellite inversions, whereas small increments appear in Northern Finland in the surface-based inversion due to the presence of the Pallas station. Despite these limitations, the comparison of satellite-based inversions with the surface-based inversion shows
some consistency: surface-based increments agree well with WFMD corrections over the Paris area and Italy, and with SRON and BLENDED corrections over the Netherlands and Switzerland. However, the surface-based inversion is not consistently closer to one satellite-based inversion than to another, as indicated by the $R^2$ correlation coefficients between surface-based increments and satellite-based increments (0.24 for SRON, 0.34 for BLENDED, and 0.37 for WFMD).

The seasonal cycle of $CH_4$ emissions is shown in Figure 10c. Satellite-based inversions exhibit similar temporal patterns,
including high emissions during December and January, a peak in April or May, and lower emissions over the summer months.



(a) Posterior - prior emissions

(b) Regional emission budgets

(c) Monthly averaged emissions

**Figure 10. (a)** Spatial distributions of the corrections to the prior $CH_4$ emissions from the inversions, in Gg/yr at the grid-cell scale (first row) and in TG/yr the national scale (second row). **(b)** Total emissions for the regions described in Table B1, in Tg/yr. **(c)** Seasonal cycle of the $CH_4$ emissions (lines) and one-sigma standard deviation (shaded area) for the entire domain in 2019, in Tg/day.

Despite these similarities, WFMD-based emissions are systematically lower than those from SRON and BLENDED, contributing to the lower annual total mentioned earlier. The monthly corrections to prior $CH_4$ emissions differ between satellite-based and surface-based inversions. For example, the peak of emissions during spring detected in the satellite inversions is not found in the surface-based inversion (Figure 10c). This peak is due to positive corrections in Western and Central Europe across all satellite-based inversions. This period coincides with a slight decline in observed $XCH_4$, and an increase (for the month of April) in the lateral boundary conditions (Figure D5). Yet, the amplitude of the emission peak raises questions about its origin.






However, both TROPOMI-based and surface-based inversions capture the summer emission minimum, which contrasts with the prior emissions, where the minimum occurs in autumn and winter. This unexpected seasonal cycle, already identified in Pison et al. (2021), seems to be related to CHIMERE transport modeling since it did not occur when using other models like FLEXPART. It is likely due to a misrepresentation of the atmospheric transport or poor representation of the seasonality of boundary conditions (Pison et al., 2021).

Moreover, the nearly identical emission increments for SRON and BLENDED (Figure 10) show that corrections for albedo and aerosol related biases in BLENDED do not lead to any significant differences of the derived emissions. Indeed, the logic for BLENDED is to solve these biases of the SRON XCH$_4$ retrievals using GOSAT retrievals. Therefore, this suggests that potential albedo and aerosol related biases (depicted in Section 3.1) are not significantly impacting the derived CH$_4$ emissions in Europe, within the scope of our study.

To complete the analysis of emission differences, a further comparison of the increments in the background components of the control vector reveals that, while the increments exhibit some similar patterns, they overall differ for the three TROPOMI products. The average background total columns are increased in average by 3.5, 1.4 and 2.1 ppb for respectively SRON, BLENDED and WFMD (Figure D4). The increments in the stratosphere share common patterns of positive increments over the Mediterranean basin, but with differing magnitudes: they are stronger for SRON and WFMD than for BLENDED. The lateral boundary increments are mainly positive over the western limit for SRON, BLENDED and the surface-based inversion, with the higher values for SRON. WFMD shows a different pattern for the increments of the lateral boundary conditions, with an increase of the background at the southern border of the domain. The time variations of the increments are overall consistent between products, only the magnitude are changed across the inversions (Figure D5).

For BLENDED, the close agreement between observed and simulated XCH$_4$ leads to a lower magnitude of both flux and background increments in comparison to other inversions. SRON has higher background increments in average than WFMD, consistently with the higher difference of observed and simulated XCH$_4$ showed in Figure 9. However, for SRON the emissions are not pulled down as strongly as for WFMD. For this latter product, the background and flux increments do not seem to be anti-correlated, which could have been expected as the differences between the observations and the prior simulations were not that high and the fluxes strongly decreased through the inversion. Therefore, the strong negative increments on the *Inv-WFMD* fluxes result of a complex balance between the local gradients of the increments on the background and on the fluxes: the system could have difficulty separating both when using the WFMD observations. The differences between products detailed in Sections 2.1.3, 3.1 and 3.2 could push the system towards different splits between background and emission optimization.

## 3.4 Separation of the causes of differences through OSSEs

Previous sections highlighted the differences in the products, in the observed and simulated XCH$_4$ distributions and in the results of inversions. To deepen the understanding of the differences of inversion outputs, here we compare the capacity of the system to improve emission estimates through the inversion, for the three products. As described in Section 2.4.3, OSSEs are a relevant tool to evaluate this capability, to perform sensitivity tests on the inversion procedure and to discriminate between the causes of the differences in the fluxes derived from various inversions. Following the method and notations of Section 2.4.3,





the "prior" refers to the perturbed prior $x^b$, while the "truth" corresponds to the unperturbed prior $x^t$. The performances of an OSSE are assessed using the relative increment, as defined in Equation 5: the closer to -100% $r$ is, the closer the posterior fluxes are to the truth (in comparison to the prior). This metric is calculated only over land to avoid averaging effects caused by weak fluxes over the sea. Table 5 shows the relative increment for the first six OSSE configurations, as well as the RMSE
and $R^2$ coefficient between the perturbed prior fluxes (averaged over all the members) and the true fluxes, and between the posterior and the true fluxes.

### 3.4.1   Reference scenarios

The *Ref-SB* and *Ref-WFMD* OSSEs demonstrate a consistent capacity to constrain emission estimates. Despite the relatively low values of $r$ (in comparison to the aim of -100%), we focus on the relative differences of $r$ between scenarios. The temporal
(Figure 11a) and spatial (Figure 11b) variations of the relative increments are consistent across the three products. WFMD presents the best performance, with a mean relative increment of -5.7%, in comparison to -3.5% for SRON/BLENDED, and a larger reduction in RMSE (Table 5). This product thus presents a higher constraint capacity in comparison to SRON and BLENDED.

| | Prior-truth | | Posterior-truth | | |
|---|---|---|---|---|---|
| ID | RMSE (g.m$^{-2}$.yr$^{-1}$) | $R^2$ | RMSE (g.m$^{-2}$.yr$^{-1}$) | $R^2$ | Relative increment |
| Ref-SB | 2.94 | 0.89 | 2.82 | 0.89 | -3.5% |
| Ref-WFMD | 2.94 | 0.89 | 2.74 | 0.90 | -5.7% |
| Ref-Surf | 2.94 | 0.89 | 2.71 | 0.90 | -7.3% |
| Common-SB | 2.94 | 0.89 | 2.82 | 0.89 | -3.5% |
| Common-WFMD | 2.94 | 0.89 | 2.77 | 0.90 | -5.1% |
| Err-SB | 2.94 | 0.89 | 2.81 | 0.89 | -3.8% |

**Table 5.** Evaluation results of 6 OSSE configurations. "Prior" refers to the perturbed prior fluxes, "Truth" to the unperturbed prior fluxes and "Posterior" to the optimized fluxes after the inversion. All the fluxes (thus the RMSE) are in units of gCH$_4$.m$^{-2}$.yr$^{-1}$. The prior and posterior have been averaged over all samples of the OSSE. The relative increment is calculated only over land. *Diff-SRON* and *Diff-WFMD* are not included since prior emissions were not perturbed. Scenario IDs refer to Table 3.

The prior seasonal variations of emissions remain largely preserved through the assimilation of pseudo-observations. Con-
sequently, the posterior time variations align closely with those of the unperturbed prior. However, the relative increment varies over the year (Figure 11a). These variations are partially influenced by the observational density, with better values occurring in summer, coinciding with an abundance of high-quality measurements, and higher (i.e. further from -100%) values in January and December, when coverage decreases due to cloud filtering. A small $|r|$ is also seen in May 2019, likely caused by the low number of observations (Figure 1b).
Regions with sparse observational coverage generally exhibit worse (i.e. closer to 0) relative increments. In particular, the system fails to improve emission estimates in Scandinavia across all three configurations (Figure 11b). This limitation is directly related to the sparse observation density in this region (Figure 1b), with nearly no observations available during winter.





Due to a sparse observation density, the system also performs poorly in the United Kingdom (Figure 11b), contributing to the poor mean relative increment over Western Europe.

Moreover, the system manages to make the posterior emissions closer to the truth in areas with high fluxes (northern Italy, Benelux, Romania) while struggling in areas with weaker signals, like Latvia and southern France (Figure 11b). This is linked to the prior error covariance $\mathbf{B}$ being proportional to the emissions (see Section 2.4.1), which limits the ability of the inversion to recover diffuse sources with low signal-to-noise ratios, as previously noted by Yu et al. (2021).

These reference OSSE scenarios demonstrate the capability of all TROPOMI products to bring emission posteriors closer
to the truth compared to the perturbed priors. The *Ref-Surf* scenario provides a better overall enhancement, with an average relative increment of -7.31%. However, this better result on average masks a high spatial heterogeneity: the relative increments are indeed very good in Western Europe and Central Europe where a number of stations are located, but fail to provide enhancements in regions with no stations, such as Spain or Romania (Figure 11b). As expected, the wider coverage of satellite observations provides the advantage of constraining the emissions on a wider area, even if there are no surface stations.

*3.4.2   Drivers of the differences of increments*

**Observation density**

The relationship between observation density and constraint potential is explored through the *Common-SB* and *Common-WFMD* OSSEs. As expected, the relative increments are slightly closer to zero as observation density decreases (Table 5). The average difference in comparison to the reference scenario is very low (less than 0.01 gCH$_4$.m$^{-2}$.yr$^{-1}$ in RMSE) for
SRON/BLENDED, with 73% of the observations kept in *Common-SB*, but slightly higher (0.03 gCH$_4$.m$^{-2}$.yr$^{-1}$ in RMSE) for WFMD because the difference in observation density is larger (45% of observations kept). The time series of *Ref* and *Common* scenarios in Figure 11a illustrate that the relative increment is overall slightly deteriorated, yet not systematically, as the number of observations decreases. The observation density is thus a driver of the potential for constraining emissions through the inversion. However it only partially explains the differences of performance between the products, since *Common-WFMD*
results in lower relative increments than *Common-SB*: even with the same number of observations, WFMD seems to show a better capability to bring emission closer to the truth. The key differences between these scenarios lie in the uncertainties associated with the measurements and the vertical parameters (such as pressure levels, AKs, and prior profiles) used to compute the satellite equivalents.

**Error rescaling**

The linear rescaling of the error associated with individual observations, described in Equation 1, increases these individual errors in the dataset of pseudo-observations. The comparison of *Err-SB* scenario to *Common-SB* shows a slight improvement of the emissions for all months (Figure 11a) and all regions (Figure 11b), when rescaling these errors. The spatial distributions are similar, as shown in Figure 11b, but with a slightly larger amplitude for *Err-SB*.

The *Err-SB* scenario is the closest SRON/BLENDED scenario to WFMD ones in terms of relative increment (Figure 11). The *Err-SB* and *Common-WFMD*, which share similar features in terms of coverage and errors, have the highest consistency





**Figure 11. (a)** Average monthly relative increment (%) over 2019 and **(b)** maps of relative increment (%), for the different OSSE scenarios. These scenarios are described in Table 3. The lower the relative increment, the more posterior fluxes are closer to the truth (Equation 5). The regions are described in Table B1.





between SRON/BLENDED and WFMD OSSEs, highlighting these features as drivers of the differences in inversion results. The remaining differences are due to the vertical parameters (pressure levels, averaging kernels, and prior profiles).

However, rescaling the observation error results in a enhancement of the emissions. This counterintuitive effect could be related to overfitting and complex indirect effects within the inversion: because of the large number of observations and the low individual observation errors, the system could struggle to overfit the observations, thus degrading the performances. This effect would be mitigated in the *Err-SB* scenario, leading to better (i.e. more negative) $r$. It can appear in the optimization process in variational inversions. In particular, at the $0.5°$ resolution, several observations constrain each component of the control vector, possibly leading to overfitting of some observations if the errors are small. Still, this effect is limited ($r$ varies by a few $0.1\%$) in comparison to the differences of $r$ between products.

**SRON-WFMD XCH$_4$ difference**

The scenarios *Diff-SRON* and *Diff-WFMD* provide insights into how the bias between products impacts the retrieved flux distribution. In this section, we do not focus anymore on the relative increment (fluxes are not perturbed in the *Diff* scenarios), but on the increments derived from the assimilation of the biased observations.

As expected, the opposite biases result in overall opposite increments. The assimilation of biased observations leads to a slight increase in total emissions for SRON (from 25.2 to 25.3 Tg/yr) and a decrease for WFMD (from 25.2 to 24.3 Tg/yr), consistently with the average positive difference between WFMD and SRON. However, these small differences in total emissions are the sum of large spatial variations, as shown in Figure 12b. The maps highlight the contribution of the WFMD-SRON difference to the SRON increments, with negative increments in Western Europe (UK, Ireland, eastern Spain), southern Italy and Austria/Czech Republic, particularly over mountainous regions like the Pyrenees and the High Tatras in Central Europe. Positive contributions are observed across most of Eastern Europe, Benelux/Germany, central Italy and the Alps. While these contributions should be closely related to the difference of XCH$_4$ columns, shown in Figure 12a, no clear correlation emerges between the bias in the concentrations and the corrections in the emissions: the average SRON-WFMD difference is generally negative and smooth, in contrast to the localized, strong increments in the inversions.

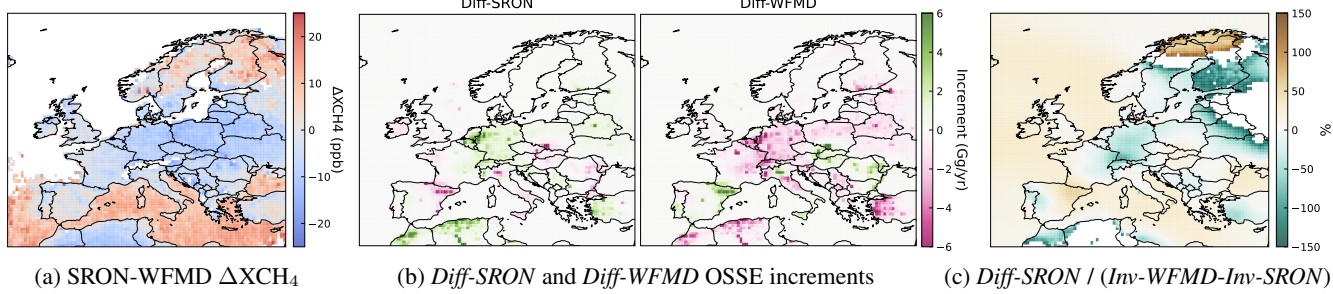

(a) SRON-WFMD $\Delta$XCH$_4$      (b) *Diff-SRON* and *Diff-WFMD* OSSE increments      (c) *Diff-SRON* / (*Inv-WFMD-Inv-SRON*)

**Figure 12.** Spatial averages of **(a)** the XCH$_4$ difference SRON-WFMD (in ppb), **(b)** the increments of the *Diff-SRON* and *Diff-WFMD* OSSEs (in Gg/yr) and **(c)** of the ratio between the increments of *Diff-SRON* by the difference of increments between inversions *Inv-WFMD* and *Inv-SRON* (in %). For the last map **(c)**, pixels with ratios above 150% (corresponding to very close increments in *Inv-WFMD* and *Inv-SRON*) were filtered out.





Yet, these two OSSE scenarios help clarify some of the patterns observed in inversions with real data. The increments of *Diff-SRON* (biased by the WFMD-SRON difference) and the difference of increments between inversions *Inv-WFMD* and *Inv-SRON* descrived in Section 3.3 are compared in Figure 12c. The ratio between these two quantities show that the differences in the inversion increments can be partially explained by the difference SRON-WFMD over the sea, in Western Europe or in Romania. However, blue areas in Central Europe and Eastern Europe show opposite variations, meaning that the OSSE increments are not sufficient to explain the increments in the inversions with real data. This suggests that while $XCH_4$ differences provide some explanation, they do not fully account for the inversion outputs. In inversions, the CAMS background is also optimized and transport is not assumed perfect, contrary to OSSEs: it makes the tracing of the main factors impacting the increments on fluxes more complex.

## 3.5 Evaluation against independent surface measurements

Finally, we evaluate the TROPOMI-based inversions described in Section 3.3 with independent surface data. To evaluate the consistency of satellite-based and surface-based perspectives, we compare the posterior simulated concentrations to independent methane observations from the surface stations listed in Table A1. The mean biases, RMSEs and correlation coefficients are summarized in Table 6.

When considering all surface measurements collectively, the differences between observations and their simulated equivalents are deteriorated for SRON and WFMD: the mean bias is respectively -9.9 and 9.1 ppb, compared to 2.0 ppb for the prior, and the RMSEs increase for both products. Nevertheless, the mean bias and RMSE are improved for BLENDED: this product is more consistent with surface station measurements than the other two. In *Inv-Surface*, the differences are decreased for almost all stations, as expected.

The overall statistics mask a heterogeneous distribution of differences across individual stations. The corrections to the prior $CH_4$ emissions derived from satellite-based inversions improve the fit with surface measurements for about half of the stations: 37% for SRON, 53% for BLENDED, 47% for WFMD (Figure 13). For SRON and BLENDED posterior simulations, the difference is deteriorated at most stations with simulated equivalents generally higher than the observations, especially over the UK, Ireland and France (MHD, RGL, TAC, SAC, OPE, Figure D2). It is consistent with the positive increments in these regions (Figure 10a). Conversely, for only 3 stations (LUT, IPR, HPB), the bias approaches zero (Figure 13). For the WFMD posterior simulation, the simulated equivalents are lower than the prior for almost all stations, due to negative flux increments in the inversions (Figure 10a). This adjustment improves the fit to surface measurements at stations mostly in Western Europe or Italy (e.g., MHD, LMP, CMN, TRN and TAC), but deteriorates the fit at most other stations (Figure 13).

These results highlight the gap between satellite-based and surface-based inversions: fitting satellite methane observations does not systematically improve the fit of the simulated $CH_4$ mixing ratios to in-situ measurements. Aligning estimates from satellite-based and surface-based inversions is crucial for ensuring consistency and reliability in methane flux estimates derived from different observational frameworks.



| Flux inputs | MB (ppb) | RMSE (ppb) | $R^2$ |
|---|---|---|---|
| Prior | 2.0 | 34.7 | 0.74 |
| Posterior SRON | -9.9 | 36.5 | 0.74 |
| Posterior BLENDED | -1.1 | 34.2 | 0.76 |
| Posterior WFMD | 9.1 | 36.7 | 0.71 |
| Posterior Surface | 1.3 | 22.3 | 0.91 |

**Table 6.** Comparison of the mean bias (MB), RMSE and $R^2$ between independent surface measurements and the simulated concentrations using respectively the prior emissions and the posterior emissions from the four inversions. The high RMSE (in comparison to the MB) highlights the variability of observation/simulation comparison for surface measurements.

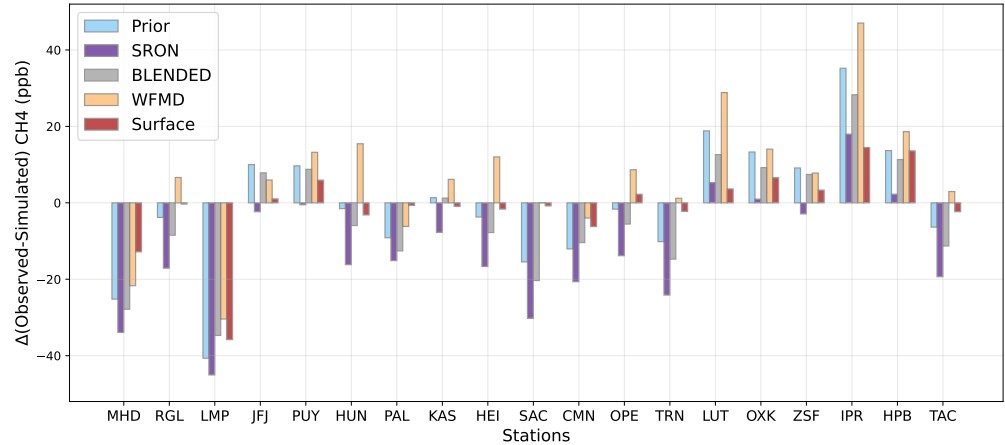

**Figure 13.** Differences between independent surface measurements and the simulations (in ppb) using the prior emissions and the posterior emissions from the SRON, BLENDED and WFMD and Surface-based inversions, for the stations described in Table A1.

## 4 Conclusions

The assimilation of $CH_4$ total columns from satellite observations into inverse modeling systems holds great promise for quan-
tifying methane emissions, as they provide complementary information to bottom-up inventories and surface measurements.
In this study, we compare three TROPOMI products, retrieved from substantially distinct algorithms, that have already been
widely used in studies from the local to the global scales. The consistency between the emissions derived from the assimilation
of the TROPOMI products is crucial for the interpretability and the validation of analyses using these datasets. The quality
filtering used here follows the official recommendations of the product providers: these filters ensure that only high-quality
data are used, maintaining a high level of coverage over Europe.

This study provides a deeper understanding of the differences between these products in terms of quality filtering, spatial
coverage and distributions, uncertainties, ability to constrain emissions and, ultimately, impact on the retrieved fluxes.





The retrievals are sensitive to a range of instrumental and atmospheric variables. A machine learning model is employed to assess the importance of features in predicting differences between the satellite products, revealing that these differences exhibit

sensitivity to factors such as aerosols, extreme albedo values, and striping patterns. The effects of these variables on the fluxes derived from inversions should be the object of further investigation. Recent and ongoing developments (e.g., reprocessing of destriped orbites for SRON, improved aerosol event filtering for WFMD, enhanced cloud filtering for both products) are expected in new product updates and should improve the quality of the products and their consistency.

OSSEs further highlight the role of both observation density and errors on the capability of the inversion system to enhance

the emissions estimates. The inter-product difference analysis pinpoints specific spatial patterns of the increments related to the SRON-WFMD XCH$_4$ difference, even if it only partially explains these patterns. The OSSEs further underscore the limitations of the inversion system and its complex dynamics. The optimization process involves a delicate balance between increments on emissions and on the background. The relative corrections on emissions and background differ across TROPOMI-based inversions, thus influencing the derived emission budgets. Furthermore, standardized observation error definitions are required.

Specifically, we recommend to rescale the observation errors for SRON and BLENDED in the case of regional inversions. A proper formula, similar to Equation 1, should be derived for each product. The existing equation 1 is already effective in rescaling the errors consistently with WFMD.

Additionally, a potential seasonal bias in the transport model warrants investigation, as it may introduce aliasing effects into the inferred corrections. Refinements in the configuration of the inversion system are thus also essential to enhance the

consistency and robustness of emission estimates.

Our findings demonstrate that assimilating the three TROPOMI products into regional inversions results in distinct posterior CH$_4$ emission estimates. Our top-down estimated CH$_4$ emissions for EU27+UK are evaluated to 25.7 Tg/yr for SRON, 25.0 Tg/yr for BLENDED and 16.9 Tg/yr for WFMD in 2019. At the monthly and sub-national (pixel) scales, the consistency between products remains insufficient for reliable budget estimates. Our study shows as expected a good agreement between

SRON and BLENDED. Since BLENDED is a post-processed version of SRON that corrects the albedo and aerosol related XCH$_4$ biases with GOSAT observations, it suggests that these biases have relatively low impact on the differences of posterior emissions between TROPOMI products. The comparison with an inversion assimilating surface station data (23.0 Tg/yr) does not conclusively indicate which TROPOMI product yields posterior emissions most consistent with surface-based estimates. Instead, the analysis underscores the difference between satellite-based and surface-based inversions, so that the choice of

product depends on the specific goals of the study, each having its own strengths, weaknesses, and sensitivities. Efforts to reconcile the surface and satellite-based perspectives are essential for improving reliability in methane emission estimates from the top-down approach.

At the regional level, higher-resolution inversions (e.g., $0.2° \times 0.2°$) hold promise, though they come at the cost of increased computational resources. Future research should extend these analyses to the global scale. Anticipated improvements in upcom-

ing TROPOMI versions (as mentionned above), coupled to improved configuration of our inversion system, should enhance the consistency between inversions, leading to more reliable CH$_4$ estimates. Another promising avenue for future work is the joint assimilation of both satellite and surface data. This option may offer a more comprehensive and accurate understanding



of methane fluxes, but requires a dedicated work on the consistency of error statistics of both observational frameworks. This progress is crucial for validating the effectiveness of European mitigation strategies and for monitoring the necessary reductions

of the $CH_4$ emissions in line with the Global Methane Pledge's 2030 reduction targets.

*Data availability.* TROPOMI $CH_4$ product (v2.4) can be found here: https://s5phub.copernicus.eu/dhus/#/home, last access May 2021 (Landgraf et al., 2024). The WFMD methane data can be accessed via http://www.iup.uni-bremen.de/carbon_ghg/products/tropomi_wfmd/, last access November 2022 (Schneising et al., 2023). Blended TROPOMI+GOSAT Satellite Data Product for Atmospheric Methane was accessed from https://registry.opendata.aws/blended-tropomi-gosat-methane, last access May 2024 (Balasus et al., 2023).

*Code and data availability.*  The CHIMERE code is available here: www.lmd.polytechnique.fr/chimere/ (Menut et al., 2013; Mailler et al., 2017). The CIF inversion system is available at: http://community-inversion.eu/ (Berchet et al., 2021).

*Author contributions.*  AB, AFC, IP and ASP contributed to the study conceptualization. AFC conducted the data collection with contribution of ASP, AM and AO; IP and ASP run the simulations. ASP conducted the analyses with contributions of AB, AFC, IP, EP, AO and GB. OS, MB, JDM and TB provided guidance on the TROPOMI data and discussed results. ASP wrote the article with input from all authors.

*Competing interests.*  The authors declare that they have no conflict of interest.

*Acknowledgements.*  A large part of the development and analysis were conducted in the frame of the H2020 VERIFY project, funded by the European Commission Horizon 2020 research and innovation programme (agreement no. 776810). This project has received funding from the European Union's Horizon Europe research and innovation programme (agreement no. 101081395). This work was also supported by the CNES (Centre National d'Etudes Spatiales), in the frame of the TOSCA ARGOS project. The last part of the analysis was conducted in

the frame of the ESA initiative SMART-CH4 (Satellite Monitoring of Atmospheric Methane, contract no. 4000142730/23/I-NS), which is part of the EC-ESA Joint Earth System Science Initiative. This work was granted access to the HPC resources of TGCC under the allocations A0140102201 made by GENCI. We thank the data providers of TROPOMI products: SRON, the Atmospheric Chemistry Modeling Group at Harvard University, and University of Bremen. We also acknowledge the principal investigators of surface stations and TCCON sites for the data used for evaluation in this work, as well as the Japanese Aerospace Exploration Agency, the National Institute for Environmental

Studies, and the Ministry of Environment for the GOSAT data.

University of Bremen acknowledges funding from the European Space Agency via project GHG-CCI+ (contract no. 4000126450/19/I-NB) and from the Bundesministerium für Bildung und Forschung within its project ITMS (grant no. 01 LK2103A). The TROPOMI/WFMD retrievals were performed on HPC facilities funded by the Deutsche Forschungsgemeinschaft (grant nos. INST 144/379-1 FUGG and INST 144/493-1 FUGG).





Finally, we wish to thank J. Bruna (LSCE) and his team for computer support and the use of the OBELIX computing facility at LSCE.



# A List of surface stations

**Table A1.** Surface stations used for the evaluation, with their coordinates.

| ID | Station | Country | Lat (°) | Lon (°) | Altitude [m a.s.l.] |
|----|---------|---------|---------|---------|---------------------|
| **Mountain** | | | | | |
| CMN | Monte Cimone | Italy | 44.17 | 10.68 | 2165 |
| HPB | Hohenpeissenberg | Germany | 47.80 | 11.02 | 934 |
| JFJ | Jungfraujoch | Switzerland | 46.55 | 7.99 | 3570 |
| KAS | Kasprowy Wierch | Poland | 49.23 | 19.98 | 1989 |
| OXK | Ochsenkopf | Germany | 50.03 | 11.81 | 1112 |
| PUY | Puy de Dôme | France | 45.77 | 2.97 | 1465 |
| ZSF | Zugspitze | Germany | 47.42 | 10.98 | 2666 |
| **Coastal** | | | | | |
| IPR | Ispra | Italy | 45.81 | 8.64 | 210 |
| LMP | Lampedusa | Italy | 35.52 | 12.63 | 45 |
| LUT | Lutjewad | Netherlands | 53.40 | 6.35 | 1 |
| MHD | Mace Head | Ireland | 53.33 | -9.90 | 8 |
| RGL | Ridge Hill | UK | 52.00 | -2.50 | 204 |
| TAC | Tacolneston | UK | 52.52 | 1.14 | 56 |
| **Other** | | | | | |
| HEI | Heidelberg | Germany | 49.42 | 8.67 | 116 |
| HUN | Hegyhátsál | Hungary | 46.96 | 16.65 | 248 |
| OPE | Obs. pérenne de l'environnement | France | 48.56 | 5.50 | 390 |
| PAL | Pallas | Finland | 67.97 | 24.12 | 565 |
| SAC | Saclay | France | 48.72 | 2.14 | 160 |
| TRN | Trainou | France | 47.96 | 2.11 | 131 |





# B  List of sub-continental regions

**Table B1.** European regions used in this study.

| Region | Countries |
|---|---|
| Western Europe | Belgium, France, Ireland, Luxembourg, Netherlands, United Kingdom |
| Central Europe | Austria, Croatia, Czech Republic, Estonia, Germany, Hungary, Latvia, Lithuania, Poland, Slovakia, Slovenia, Switzerland |
| Southern Europe | Italy, Portugal, Spain |
| Northern Europe | Denmark, Finland, Norway, Sweden |
| South-Eastern Europe | Albania, Bosnia-Herzegovina, Bulgaria, Cyprus, Greece, Macedonia, Moldova, Montenegro, Romania, Serbia |



## C  Comparison to TCCON observations

In addition to the comparison of the XCH$_4$ distributions of the TROPOMI products, we compare the TROPOMI-TCCON co-
located observations for the seven TCCON stations listed in Table C1. The Total Carbon Column Observing Network (TCCON)
is a network of ground-based stations equipped with similar high-resolution spectrometers (Bruker IFS) and using a common
retrieval algorithm to ensure comparability of the measurements. The network consists of 28 operational sites, of which 7 are
in the domain of this study. It is available at https://tccondata.org/. We use the last update of GGG2020 (Laughner et al., 2024).
Previous studies have compared one or two TROPOMI products to TCCON observations (T. Hilbig et al., 2023; Balasus et al.,
2023; Borsdorff et al., 2024; Lindqvist et al., 2024), but none of them have directly compared the 3 products all together.

| ID | Station | Country | Lat (°) | Lon (°) | Altitude [m a.s.l.] |
|-----|---------|---------|---------|---------|---------------------|
| BRE | Bremen | Germany | 53.1 | 8.85 | 30 |
| GAR | Garmisch | Germany | 47.48 | 11.06 | 745 |
| KRL | Karlsruhe | Germany | 49.1 | 8.44 | 110 |
| NIC | Nicosia | Cyprus | 35.14 | 33.38 | 185 |
| ORL | Orléans | France | 47.96 | 2.11 | 130 |
| PAR | Paris | France | 48.85 | 2.36 | 60 |
| SOD | Sodankylä | Finland | 67.37 | 26.63 | 188 |

**Table C1.** TCCON stations used for the evaluation.

To compare observed CH$_4$ total columns from TROPOMI and TCCON datasets, we consider the co-located observations that
are within 1 h and 100 km of each other, with a maximum surface elevation difference of 250 m. For co-located observations,
it is required to adjust the columns for the differences of vertical sensitivities and prior XCH$_4$ profiles used in the retrievals,
using the TCCON profile as the common prior profile. Following Apituley et al. (2024), Schneising (2022) and Balasus et al.
(2023), the vertical profiles of TCCON (51 levels) are interpolated on the TROPOMI layers $l$ (20 layers for WFMD, 12 for
SRON and BLENDED). The adjusted TROPOMI XCH$_4$ total column $\hat{y}_{adj}$ is thus, with $\boldsymbol{y}_{a,TC}$ and $\boldsymbol{y}_{a,TR}$ the TCCON and
TROPOMI prior profiles, $\hat{y}$ the TROPOMI XCH$_4$ total column and $\boldsymbol{a}$ the column averaging kernel:

$$\hat{y}_{adj} = \hat{y} + \sum_l h^l (1 - a^l)(y_{a,TC}^l - y_{a,TR}^l) \tag{C1}$$

The results of the comparison for 2019 are presented in Table C2 and Figure C1. WFMD tends to overestimate methane
concentrations, SRON has the lowest mean of the daily averaged difference but higher deviations and lower correlation in
comparison to the other products. BLENDED observations align more closely with TCCON in terms of $R^2$ and RMSE, with
a negative offset that is rather uniform across the stations. The average for individual stations are consistent with Balasus et al.
(2023) and T. Hilbig et al. (2023) for SRON and BLENDED. However, they differ from the results of Borsdorff et al. (2024)
and Lindqvist et al. (2024) for WFMD and from the results of T. Hilbig et al. (2023) for SRON. Overall, the values of the
differences between TROPOMI and TCCON XCH$_4$ fall in the range [-25,+25] ppb. Moreover, the relative accuracy (standard



deviation of the mean local offsets relative to TCCON at the individual sites) of TROPOMI products shown in Table C2 are below the 10 ppb threshold deemed suitable for regional inversions by Buchwitz (2015). BLENDED and WFMD have lower relative accuracies (3.1 and 3.3 ppb) than SRON (4.8 ppb).

| Product | Mean (ppb) | $R^2$ | RMSE (ppb) | Relative accuracy (ppb) |
|---------|-----------|-------|-----------|-------------------------|
| SRON | 0.9 | 0.48 | 16.4 | 4.8 |
| BLENDED | -2.6 | 0.62 | 11.1 | 3.1 |
| WFMD | 7.5 | 0.54 | 13.7 | 3.3 |

**Table C2.** Mean and RMSE of the daily averaged difference TROPOMI-TCCON, as well as the correlation ($R^2$) and the relative accuracy of the TROPOMI products relative to TCCON, over the co-located observations at the TCCON stations in 2019. The relative accuracy is the standard deviation of the mean local offsets relative to TCCON at the individual sites.

Analysis of individual stations reveals similar patterns for those located in Western Europe (Bremen, Karlsruhe, Orléans and Paris). For these stations, SRON and WFMD show comparable distributions (with WFMD values slightly higher), while BLENDED systematically produces lower median values, consistently with the comparison of XCH$_4$ distributions detailed in Section 3.1. A similar pattern can be seen in Nicosia, except for SRON higher values. All the products have a similar positive offset in Garmisch in comparison to other Western Europe stations. For this station, located in the Northern Alps, the offset is likely due to a bias associated with albedo or difference in the altitude of the ground pixel of the satellite and the station. Time series (not shown) indicate a very low seasonal dependency in the differences, apart from Sodankylä. In this high-latitude station in Finland, the bias in 2019 is positive during spring, and negative in autumn, consistent with the findings of Lindqvist et al. (2024). This seasonal variation is only present in 2019. Due to this temporal variations and to the limited number of co-located observations at this latitude, the deviations of the TROPOMI-TCCON differences are amplified, especially for SRON which has the largest seasonal variations. These results highlight the challenges of using TROPOMI at high latitudes, where coverage is sparse, and uncertainties are large.

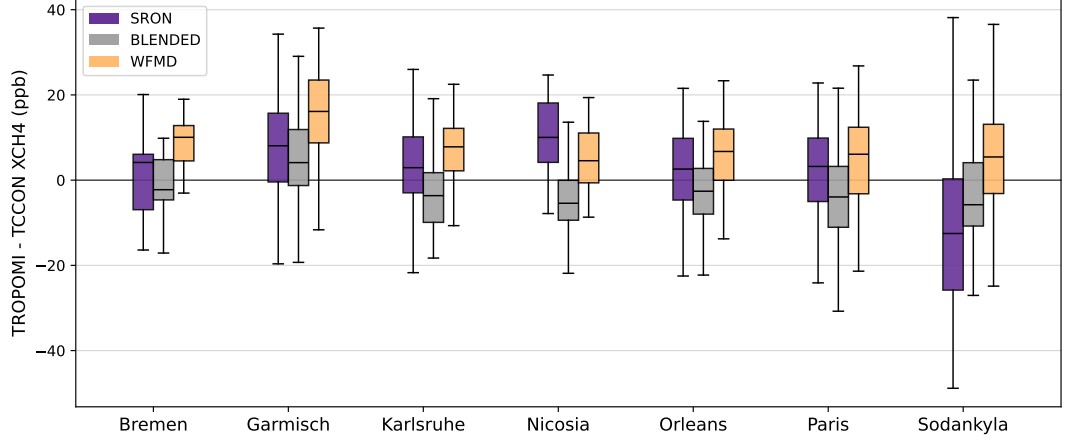

**Figure C1.** Median and quartiles of the daily averaged differences between TROPOMI and TCCON XCH$_4$ (ppb) for each station selected for the evaluation, in 2019.





# D    Additional figures

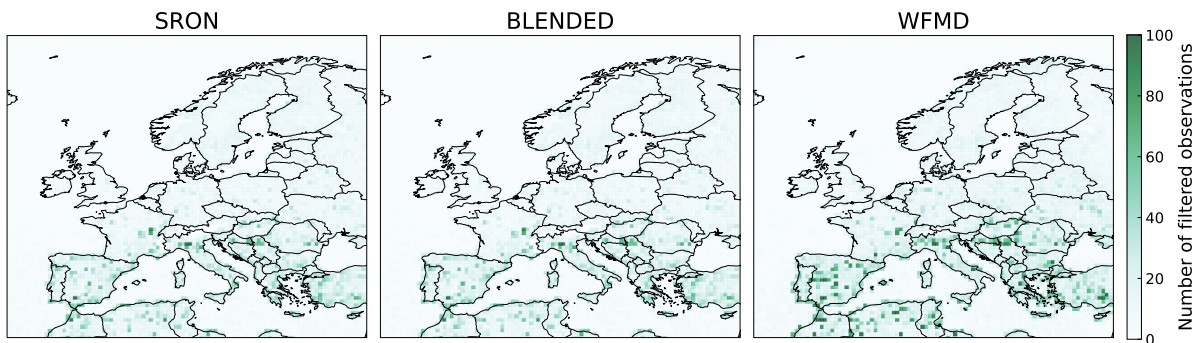

**Figure D1.** Count of observations that have been filtered in the post-processing of the TROPOMI products, as described in Section 2.4.2

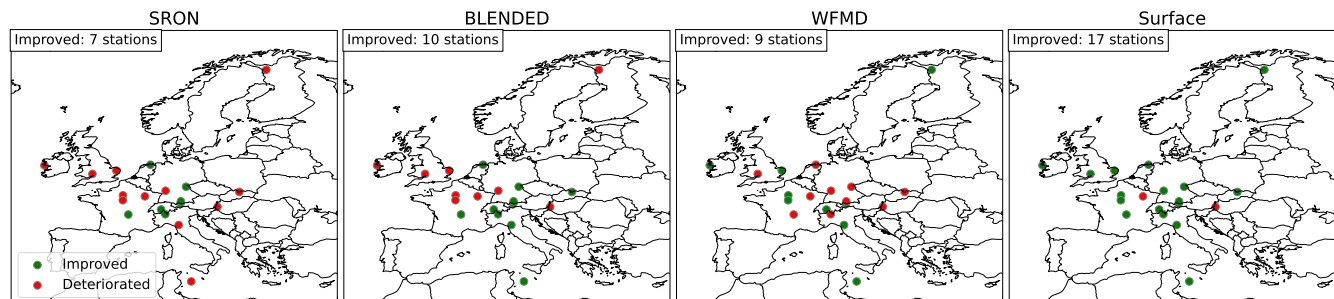

**Figure D2.** Evaluation map of the satellite-based and surface-based inversions: green (resp. red) circles are the surface stations for which the posterior simulated concentrations are in average closer (resp. further away) to the observations than the prior ones.



**Figure D3.** Seasonal average difference between TROPOMI observed concentration and CHIMERE simulated equivalent. MB is the mean bias and RMSE the root mean square error. Units are pbb. JFM, AMJ, JAS and OND are acronyms referring to the months of each season.



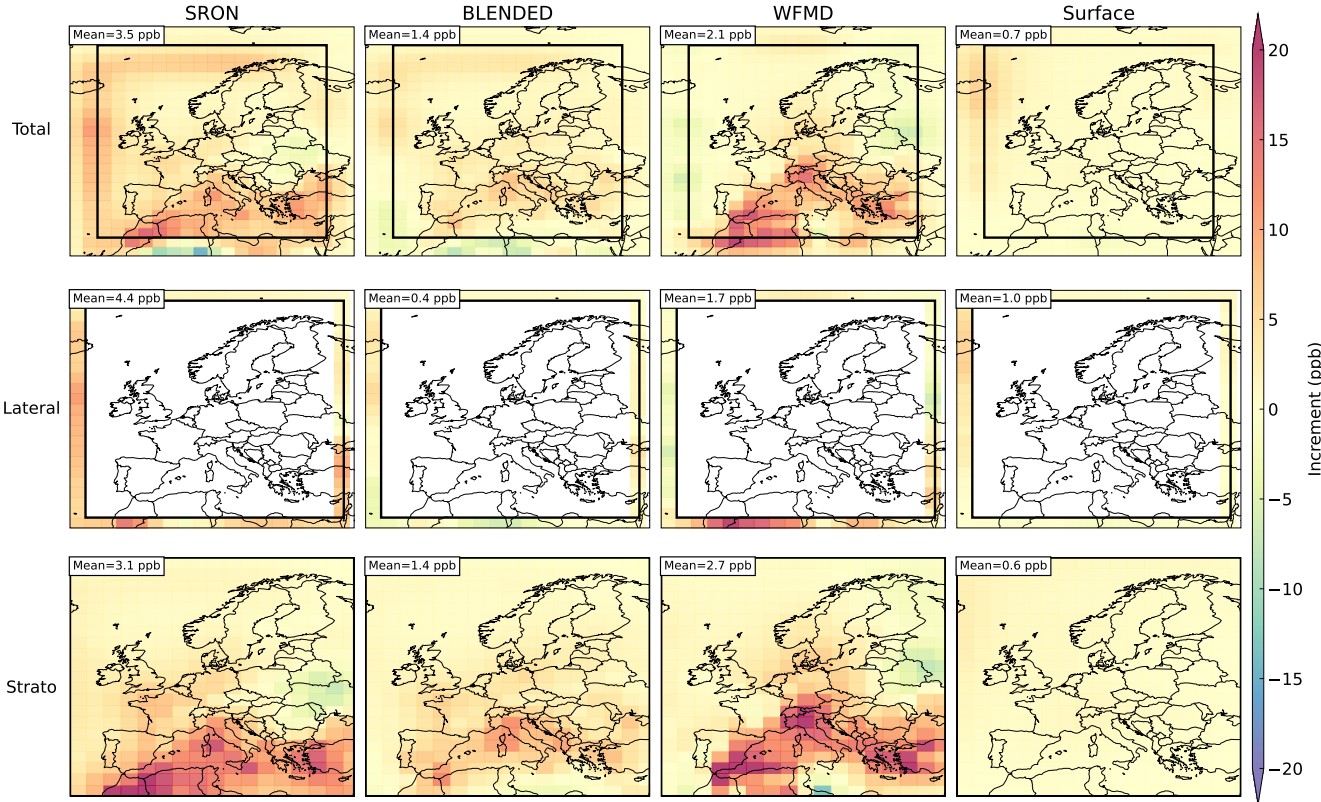

**Figure D4.** Increments of the components of the background, for the 4 inversions presented in Section 2.4.2. The first row shows averaged total columns, the second and third row only the pixels used as lateral and top boundary conditions, and the fourth row shows the average stratosphere column (for pressures > 200 hPa).

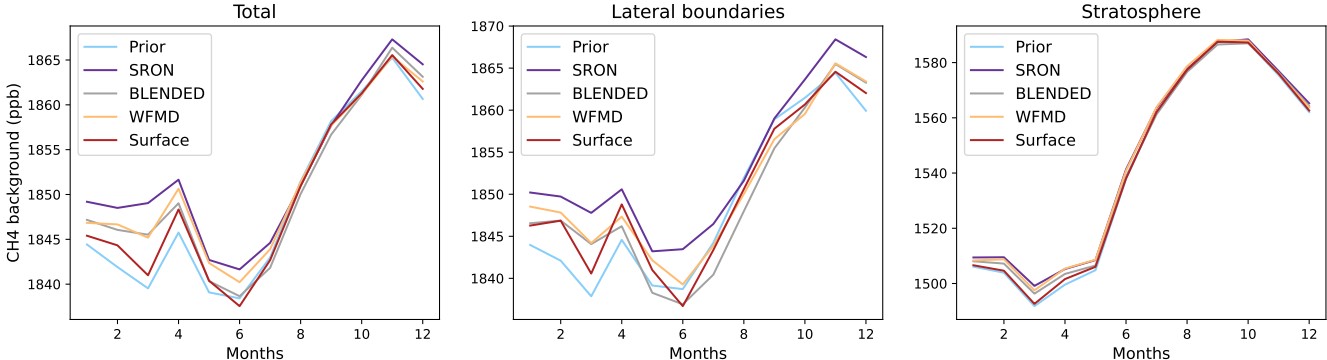

**Figure D5.** Time series of monthly averaged increments of the components of the background, for the 4 inversions presented in Section 2.4.2. The components of the background are the same as in Figure D4.



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
