# Peer review of "Can we obtain consistent estimates of the emissions in Europe from three different CH4 TROPOMI products?"

_EGUsphere, 2025_

## Referee Comment (RC2)

This manuscript by Sicsik-Paré et al. presents a comparative analysis of three XCH$_4$ products derived from TROPOMI observations—SRON, BLENDED, and WFMD. The authors examine various parameters influencing discrepancies among the XCH$_4$ datasets, highlighting aerosol scattering and SWIR albedo as primary contributors. Additionally, the study utilizes these products in inverse modelling with the CHIMERE model and a data assimilation system to estimate surface methane fluxes.

Overall, the manuscript provides valuable insights for the atmospheric methane community, especially in the context of inverse modelling and satellite product evaluation. It demonstrates the potential and limitations of different XCH$_4$ retrievals and their influence on flux estimates. I recommend publication in ACP after addressing the following general and specific comments.

**General comments:**

1. The manuscript is generally well-structured but somewhat lengthy in certain sections. For instance, the detailed description of the CHIMERE model in Section 2 could be moved to the appendix to improve the flow and readability of the main text.

2. The WFMD product reports the highest absolute XCH$_4$ values among the three datasets and shows the largest bias relative to TCCON (7.5 ppb; see Table C2). While the WFMD XCH$_4$ dataset shows moderate differences from both prior and posterior simulations compared to SRON and BLENDED, the fluxes derived from WFMD are consistently the lowest (Figure 10b). This discrepancy raises discussion here.

Could the lower flux estimates be a result of overestimated background subtraction? Furthermore, WFMD includes approximately 30% more retrievals than the other products. Could this may be due to less stringently filtered, allowing residual uncertainties to remain, which could contribute to the observed differences in flux estimates?

3. While the study uses surface observations for the inversion, have the authors considered assimilating ground-based total column data, such as from TCCON or COCCON? These measurements are widely used to validate satellite products, as both observe total atmospheric columns and are therefore directly comparable. Incorporating such datasets could provide a more representative constraint on regional methane emissions and may help improve flux estimates, particularly in regions where surface observations are sparse or not fully representative.

**Specific and technical comments:**

Line 126-127: the authors reference previous work to justify scaling the errors by a factor of 2, but a brief explanation here would enhance clarity for readers unfamiliar with that context.

Line 207: Figure 3 is referenced before Figure 2 in the text. The order of figures should be revised for logical flow.

Line 207-208: The phrase "BLENDED lower concentrations" is unclear. Does this mean that only the lower end of BLENDED values agrees well with GOSAT? Consider clarifying this point. Additionally, including GOSAT's spatial distribution or mean value in Figure 3a would improve visualization and support the comparison.

Figure 3: It seems the errors for SRON and BLENDED largely overlap in panels (b) and (c). If this is the case, please clarify it in the figure caption.

Line 240: typical of SWIR retrievals >>> type?

Line 408: The analysis indicates that differences are closely linked to across-track pixel index, aerosols, and SWIR albedo. While the latter two are physically intuitive, the role of across-track pixel index is less clear, as it represents only a positional identifier. Could these index be indirectly capturing information related to albedo, aerosol, or stripping artifacts in the retrievals? Further clarification would be beneficial.

Line 519: Background removal is known to be challenging. The manuscript would benefit from a brief explanation of how the background is defined and removed in this study.

Line 531-532: result of >>> result from?

Line 560: The authors note poor improvement over Scandinavia. Could extending the study period improve coverage and reduce uncertainty through increased data availability? This point deserves further discussion.

Appendix C: COCCON is increasingly used as a complementary dataset to TCCON, particularly in regions lacking TCCON coverage. The authors could also consider incorporating COCCON measurements to provide additional reference data for satellite validation, thereby enhancing the spatial representativeness of the evaluation.

---

## Referee Comment (RC3)

**Peer Review Report**

*Title: Can we obtain consistent estimates of the emissions in Europe from three different CH$_4$ TROPOMI products?*

**General Assessment**

This manuscript presents a comprehensive study of methane (CH$_4$) emissions over Europe using three different satellite-based TROPOMI products. The central question—whether consistent emission estimates can be derived from these different products—is of high scientific and practical relevance. The increasing reliance on satellite-based greenhouse gas observations for policy-making and climate monitoring underscores the importance of this work.

The authors have conducted a broad and technically rich analysis, including detailed product processing, uncertainty treatment, and inverse modeling. However, the manuscript currently lacks a coherent narrative that leads the reader from the stated question to a clear, data-driven conclusion. While the technical sections are sound, the structure often obscures the main message. There is an overemphasis on methodological detail and extensive referencing at the expense of direct interpretation and hypothesis testing.

The authors need to significantly improve the clarity, structure, and focus of the manuscript to fulfill its potential. In particular, the main hypothesis must be explicitly stated, tested, and either confirmed or rejected using statistical evidence. A clearer distinction between consistency and reliability (e.g., agreement with ground truth) should also be maintained throughout.

**Major Comments**

**1. Clarity of Research Question and Structure**

The central question posed in the title is not clearly answered in either the abstract or the conclusions. A well-defined hypothesis and its statistical verification should guide the structure of the manuscript. Much of the methodology section feels detached from the central question. The authors should restructure the manuscript to follow a logical flow: problem statement → methodology → analysis → results → conclusion.

**2. Abstract and Introduction**

The abstract is vague and at times contradictory. Replace vague statements with a direct summary of findings and implications. The introduction includes redundant background. Replace with targeted discussion of satellite CH$_4$ retrievals, the TROPOMI product landscape, and the importance of consistency.

**3. Definition of Consistency and Its Measurement**

Clearly define how consistency is measured (e.g., correlation, bias, RMSD). Introduce the statistical metrics and how they will be applied across spatial scales.

**4. Overemphasis on Methodological Details**

Trim excessive detail from Sections 2.1–2.4. Move technical descriptions to the Supplement unless they directly inform the consistency analysis.

**5. Statistical Analysis and Uncertainty**

Introduce proper statistical hypothesis testing and include uncertainty bounds, p-values, or confidence intervals to assess significance.

**6. Use of Single-Year Data**

Explain why 2019 is used. A single year may not capture variability. Consider a multi-year analysis or justify why this year is representative.

**Minor and Editorial Comments**

**Abstract**

- Improve focus and clarity. Avoid vague phrases such as 'holds promise' or 'paving the way.'
- Quantify findings where possible.
- Clarify ambiguous phrases (e.g., 'relative increase' of what? Compared to what baseline?).

**Introduction**

- Avoid textbook-level background on methane (lines 17–24).
- Clarify references to BU/TD inventories—if not directly used, they may be omitted.
- Include a brief overview of available $XCH_4$ products and their known limitations.
- Define technical terms when first introduced.
- Clarify why Europe and the year 2019 were chosen.

**Methods**

- Remove unnecessary numeric details unless they support later discussion.
- Equation 1: Define all variables and explain relevance.
- Explain key terms such as 'pseudo noise,' 'observational error,' 'qa_value.'
- Provide a transparent uncertainty treatment methodology.

**Results**

- Focus more on direct comparisons of the three products.
- Avoid subjective or speculative language.
- Provide uncertainty estimates for seasonal/annual budgets.

- Correct figure references (e.g., D4/D5).

**Discussion and Conclusions**
- Answer the title question clearly and directly.
- Support general claims with quantitative or literature-based evidence.
- Clarify how data quality and coverage interact.
- Specify the impact of individual factors on retrieved fluxes.

**Recommendation**
Minor Revision

This manuscript addresses an important question with strong technical depth, but the central message is lost in overly detailed methodology and a lack of structured hypothesis testing. I encourage the authors to revise the manuscript with a clearer focus on answering the title question, supported by statistical analysis, direct comparisons, and more effective framing of results and conclusions.

---

## Author Comment (AC1)

**Author response – RC1**

We thank the reviewer for their thoughtful comments. We respond to them below: the comments are copied hereafter and shown in black, our author responses in blue; suggested new manuscript text is indicated in green. New line numbers in the revised manuscript are provided.

A comparison is presented of three methane (CH4) retrieval products for the TROPOMI instrument. Inversions are performed using the TROPOMI data, the CHIMERE model and a variational data assimilation system. Substantial differences in European fluxes were derived for the inversions using the three TROPOMI retrievals (SRON, BLENDED, WFMD) and these each differed from inversions using the surface data. This result has important implications for the use of TROPOMI data in regional methane inverse modelling.

Overall, I find the study to be important, timely and thorough. However, I feel that the structure of the paper could be improved for clarity and brevity. In particular, I would urge the authors to consider the following:

- 1. Are all OSSEs strictly necessary? In particular, I wondered if the "diff" OSSEs could be cut (or at least moved to the Appendix/Supplement), without detriment to the paper.
- 2. In many places, the structure and text could be improved for readability (see suggestions below).

In this study, the OSSEs are the key method to understand the drivers of the differences in emission estimates between inversions. The « *Diff* » OSSE scenarios address the impact of the differences in XCH4 spatial and temporal distributions on retrieved emissions: going further than the random perturbations in other OSSEs, they allow to evaluate the impact of bias between observation datasets. For this reason, we have chosen to keep this part. In the conclusion, we highlight the value of these simulations.

Regarding the structure, we have made changes (in particular in the abstract, Section 2 and the conclusion) to make the text more concise. The structure has been adapted (e.g., the considerations on observation errors in Section 2 were grouped in 2.1.3) and some elements that were not necessary for the understanding of the results have been removed. We hope that these changes will clarify the key statements of the article and improve readability.

**General comments**

1. I don't understand why the WFMD product leads to substantially lower fluxes than the other inversions. The terrestrial mole fractions seem to be between those of SRON and BLENDED (Figure 9), and the lateral boundary conditions seem to be similar to BLENDED (D5). The authors address this on L532: "Therefore, the strong negative increments on the Inv-WFMD fluxes result of a complex balance between the local gradients of the increments on the background and on the fluxes: the system could have difficulty separating both when using the WFMD observations." But can it really be that complicated? If the boundary conditions are roughly similar between two products, but the terrestrial mole fractions are mostly higher for one (Figure 9), then surely, the fluxes for that product must be higher, not lower? Furthermore, it's not clear why their explanation (that the fluxes and

background can't be easily separated) would only apply to WFMD. I wonder if there could be a bug here...

The lower emission estimates derived from the WFMD inversion are indeed a crucial element of analysis in this study. This element is discussed at the end of Section 3.3. We did not manage to identify a unique origin of these strong negative increments: the distribution of obs-sim is overall positive (Fig. 9) but with values closer to 0 than SRON, so we expect lighter increments than for SRON inversion – this is not the case. We investigated the background optimization (strong negative increments could have compensated strong positive increments on the background, which is not the case) but it was not conclusive either (Fig. D4, D5).

This is likely due to a subtle balance within the inversion process: the differences in spatial distributions of background increments and emission increments with the other products could come from differences in the separation of background and emissions when using WFMD observations. The high number of retrievals could also result in overfitting in the optimization process, and/or residual XCH4 uncertainties due to less stringent filtering. This topic has to be further explored to allow better understanding of the process.

The results of Rona Thompson presented at EGU 2025 also showed similar relative differences between products (Prior: 27.4 Tg/yr, SRON: 26.4 Tg/yr, BLENDED: 23.7 Tg/yr, WFMD: 19.0 Tg/yr, ICOS: 22.9 Tg/yr). These simulations were made with another model (FLEXPART). It cannot be considered a strict validation, but it suggests that WFMD observations do indicate lower CH4 emissions than the other products in Europe in 2019.

Finally, the structure of the Community Inversion Framework (CIF) makes unlikely the occurrence of a bug, as all the inversions are processed with the exact same code and processing.

Reference: Thompson, R., Schneider, P., and Stebel, K.: Using different TROPOMI XCH4 retrieval products in atmospheric inversions of CH4: a comparison and reconciliation over Europe, EGU General Assembly 2025, Vienna, Austria, 27 Apr–2 May 2025, EGU25-9567, https://doi.org/10.5194/egusphere-egu25-9567, 2025.

2. Throughout, why are posterior flux uncertainties not provided, except for in Figure 9 monthly means?

In the framework of 4D-Var inversions, there is no direct calculation of the posterior uncertainty. It has to be made separately from inversions as such. This is a limitation of our approach, as the comparison of emission estimates between several inversions requires the comparison of uncertainty ranges.

Ensemble methods are often used for this purpose, but with a high computational cost. Following this approach, the uncertainty reduction (independent of the observation vector) can be estimated through OSSEs: computing the standard deviation of priors and the one of posterior emissions across the ensemble samples, the ratio of the two gives the uncertainty reduction. For the total budgets, it is estimated to 78% reduction for SRON and BLENDED, 74% for WFMD and 51% for surface stations. However, we do not use these values for setting error bars to our budget estimates, for two reasons: 1) the poor statistics of this ensemble (4 samples) and 2) the lack of proper uncertainty estimation for the prior as it is not provided with the emission inventories.

Figure 9 contains 1-σ ranges, which are not proper uncertainties: they are deviation of the weekly fluxes used for the monthly average. Clarification has been added in the legend of the figure.

**Specific comments**

Many statements in the abstract are ill defined, or vague:

L 7-8: it's not stated what these increases or decreases are relative to

The increments are relative to the prior, it has been added in the text.

L 8: "Seasonal emissions are highly correlated across the inversions." I'm not really sure what the point of this sentence is, or what it means. Cut?

The sentence was not clear and therefore removed.

L10: What does it mean that the boundary conditions differ "substantially" for WFMD? Also, is this true? It doesn't seem so from Figure D5.

The sentence was modified, as it was confusely mixing increments on the background and on emissions. The comments on background have been removed of the abstract as they are not essential for the main message of the article.

L10: "Evaluation with independent surface stations shows error reduction for about half of the sites, with BLENDED performing best". I think this means that the residual between the observations and the posterior mole fractions is reduced for about 50% of the monitoring sites. But BLENDED showed a reduction in residual for more sites than the other inversions? More precise language is needed.

This statement was rephrased and quantified to enhance clarity: « Evaluation with independent surface stations shows that the residual between the observations and the posterior concentrations is reduced for 37%, 53% and 47% of the stations, respectively, for SRON, BLENDED and WFMD ».

L11: "However, no product is systematically closer to the emissions estimated when assimilating surface observations". This is too subjective a statement. In any case, to me, the SRON and BLENDED inversions looked very similar to the surface inversion for  $\sim$ 10 months of the year, whereas WFMD differs from the surface inversion most of the time.

This sentence was changed to « No inversion provides a systematically closer match to the spatial distribution of emissions derived from surface observations. » The statement focuses on the spatial distributions of emissions.

L13 and L14: What "errors" and "quality filters" are being referred to here?

These mentions have been removed. We refer to « coverage » and « individual observation error » in the context of OSSE to make the sentence clearer.

**Main text**

L22: "with higher rates OF INCREASE over the..."

This was corrected, but the sentence has been removed for concision.

L39: "relative", rather than "relatively"

The change has been done.

L45: Shouldn't a range of wavelengths be provided ("spectral range")

The sentence has been removed for concision, the SWIR wavelength range of TROPOMI is provided in Section 2.1.1 (L.96).

L124: Briefly (1 or 2 lines) describe the de-striping procedure

The procedure is now described: « This empirical approach consists in removing the CH4 background by a median smoothing in the cross-track direction, and then computing a per orbit stripe value as a median in the flight direction, which is used for correction».

L137: We keep only THE highest quality...

The change has been done.

Equation 1: define sigma and sigma\_hat

Equation 1 and other considerations on the definition of errors have been grouped in Section 2.1.3 (L.195-214). Sigma and sigma\_hat are now defined in the text (L.204-207).

L164: Couldn't it be confusing to use  $\Delta x$  here? It could imply only in one horizontal coordinate. Just say within 0.01 degree lat/lon?

The description has been removed for concision and to avoid confusion, as suggested. We only refer to « common » observations, which we think is clear enough.

L180: This is the first reference to a figure (Figure 5). I presume the journal will require that figures are referenced in order?

The reference has been removed to ensure figures are referenced in order.

L208: Why is this interesting? It doesn't actually say, but seems to be implying something. Is this sentence needed?

This sentence was indeed confusing. It has been removed.

L215: "e.g. in Scandinavia". Be more specific: what are the patterns in this region.

The relative comparison of the patterns in Scandinavia has been added: « higher XCH4 than SRON for WFMD and lower for BLENDED, as seen in Figure 2 ».

L216 "The temporal variations... show consistent patterns across the products and align rather well with GOSAT". What is the basis of this statement? To me, GOSAT looks very different to SRON and WFMD, but similar to BLENDED.

The analysis of temporal variations has been improved (L.179-187). Differences were quantified and the comparison with GOSAT has been clarified. In the initial version, authors wanted to compare relative variations, but the « relative » was not clearly mentioned, making the statement confusing. It has been changed in the new version.

L234: Is this really an order of magnitude? Isn't it about a factor of 3?

The change has been done.

L240: I don't understand how the 0-5% difference is quantified to a profile. Is this per level?

It is a per level difference. This has been clarified in the text: « The per level relative difference between SRON/BLENDED and WFMD vertical profiles is below 5% for all levels. »

L325: Do you really mean "surface roughness" here? If so, you could use surface roughness (the meteorological term) as a filter, rather than topography...

The use of « surface roughness » has been changed to « subpixel topographic variability ».

L390: If the dataset is split randomly, isn't there going to be substantial correlation between the testing and training sets that influences the metrics? Most of the testing set will be adjacent to points that have been used in the training. It would be preferable to have the testing and training set be separated in space/time (or some other factor).

The random split of the dataset can indeed generate correlation between the training and testing sets. A sensitivity test was run using a different split for the SRON-WFMD pair, with Jan. to Oct. 2019 as the training dataset and Nov./Dec. 20219 as the testing set. The performance of the prediction was slightly lower, with RMSE of 8.9 ppb (in comparison to 8.1 ppb with the random split) and R2 of 0.51 instead of 0.58.

However, we are not directly interested in the performances of the model but rather in the feature contributions in the calculation of the prediction. Indeed, the SHAP value analysis was very similar in both cases, with differences of average |SHAP| (Fig. 7) between both cases inferior to 0.1 ppb. To avoid seasonal bias that could be introduced by the temporal split of the data, we chose to keep the random split of training/testing set.

Figure 7 caption: this figure needs explaining more thoroughly

Explanations have been added to facilitate the reading of the plot.

L413: since this is the first line of a paragraph, restate this initial sentence so that it reminds the reader what you're talking about.

The reminder has been added.

L450 and elsewhere: try to avoid subjective terms like "best".

The change has been done here and for other occurences of such subjective words.

L472: You can't say that the inversion "correctly" fits the data as there will always be errors. Just say that the residual is reduced after the inversion (which it must be, if the inversion is working correctly).

The sentence has been restated accordingly: « the residual (difference between the observed XCH4 and the posterior simulations) is reduced after the inversion ».

Figure 10: why is this the only place where emissions uncertainties are provided?

See answer to General comment #2.

L506: "Yet, the amplitude of the emission peak raises questions about its origin." This sentence implies something but doesn't spell it out. What are the questions, what could be the origin? Or do you just mean that you don't believe that this peak is real? If so, say so, and justify your reasoning.

The sentence has been changed to **«Yet, the origin of this emission peak has not been clarified.»**. We investigate the origin of the peak but could not find a process that fully explains it. The elements cited just before, the slight decline in observed XCH4 and the increase in the lateral boundary conditions in April, are remarkable elements but do not provide a clear explanation. Therefore, to avoid unclear interpretations, we only mention that we cannot certify what is causing this peak.

L507, L512 and elsewhere, don't start paragraphs with "However,", "Moreover", etc. Each paragraph should tackle a single idea. These words imply a continuation of an idea.

The change has been done here (L.468, 473) and for other occurences (L.526, 560).

L556: Avoid "better"

The change has been done here (L.517) and for other occurences of such subjective words.

L602: "It can appear in the optimization process in variational inversions." This is a sentence fragment. Reword.

This sentence is not necessary for the explanation, it has been removed.

L606 – 629: Could this section be removed?

See answer to General comments.

L635 – 637: I don't know what this sentence means. Rewording is needed for clarity.

This sentence has been rephrased (L.595). Here we compare the difference between observed and simulated XCH₄ based on: 1) the prior emissions 2) the posterior emissions estimated for the inversions of each TROPOMI product.

For SRON and WFMD, the average difference obs-sim and RMSE are higher (in absolute values) in comparison to the prior ones. For BLENDED, they are lower (in absolute values).

**Conclusions**: I think the conclusions are far too long and contain several statements that aren't really justified, given the results. The authors should refocus this section for concision.

The conclusion has been the object of a concision and rephrasing work, to refocus the main message of the article and to avoid making confusing statements that are not well supported by the results.

L661 – 662: This one-sentence paragraph doesn't seem necessary for a conclusions section. It doesn't really say anything.

This sentence has been removed for concision.

L675-676: What is meant by a "proper formula"? Wouldn't it be better to say what is physically needed here. I.e., how should the error be appropriately calculated.

We have described more thoroughly the calculation of the error that we recommend (L.641-644): we suggest a linear regression of the scatter of the observations relative to TCCON to derive the individual observational error, similar to what is done for the WFMD product (Eq. 1).

L678: Are model biases really a key outcome of this study? Do we need to state this here?

The model error was not directly studied. This study identified issues that could be related to issues in the transport modeling part of the simulation (reversed seasonal cycle of CH4 emissions, peak of emissions in April/May...). But there was no thorough analysis of them, they are just some leads to interpret the results. Therefore, we have removed this sentence to avoid confusion.

L679: "Refinements in the configuration of the inversion system are thus also essential to enhance the consistency and robustness of emission estimates." What does this mean? Why is it essential? What about your results indicates this to be true? (If you are just saying that inverse modelling systems need to be improved, you can cut this, as it's well understood, and you don't address this in your paper).

Indeed, improvements of transport and inverse modelling were not addressed in this study. The sentence has been removed.

L693 – 694: Your results don't show why we need higher resolution, and, clearly, global inversions would be preferable. You can safely cut these lines.

The sentence has been removed.

L697: Does your work really suggest that joint in situ and TROPOMI inversions would improve matters? At the moment, you show major differences that are difficult to reconcile. Perhaps, if we knew how to appropriately reconcile these systematic differences. But at the moment, I wonder if your work shows that in fact, we're not ready yet?

Sections 3.3 and 3.5 highlight differences between results of the in situ and satellite inversions. Reconciling the two types of inversions is a great perspective, but indeed this work indicates that it is still a challenging problem. The sentence is more of a long-term perspective than a conclusion of this study, we have removed it from this article.

**Supplementary material**

Figure D4: Is this missing a 4th row (mentioned in the caption)?

The caption has been updated. The 4th row existed in a previous version, but was removed later. Also, this figure is now D5, to ensure the correct order of figure referencing.

---

## Author Comment (AC2)

**Author response - RC2**

We thank the reviewer for their thoughtful comments. We respond to them below: the comments are copied hereafter and shown in black, our author responses in blue; suggested new manuscript text is indicated in green. New line numbers in the revised manuscript are provided.

This manuscript by Sicsik-Paré et al. presents a comparative analysis of three XCH8 products derived from TROPOMI observations—SRON, BLENDED, and WFMD. The authors examine various parameters influencing discrepancies among the XCH8 datasets, highlighting aerosol scattering and SWIR albedo as primary contributors. Additionally, the study utilizes these products in inverse modelling with the CHIMERE model and a data assimilation system to estimate surface methane fluxes.

Overall, the manuscript provides valuable insights for the atmospheric methane community, especially in the context of inverse modelling and satellite product evaluation. It demonstrates the potential and limitations of different XCH8 retrievals and their influence on flux estimates. I recommend publication in ACP after addressing the following general and specific comments.

**General comments:**

1. The manuscript is generally well-structured but somewhat lengthy in certain sections. For instance, the detailed description of the CHIMERE model in Section 2 could be moved to the appendix to improve the flow and readability of the main text.

The feedbacks from the different reviewers all suggested to improve concision, especially in Section 2. This section was reorganized to remove unnecessary statements that did not support the main message of the article. In particular, we have made the description of CHIMERE more concise. However, we prefer to keep this subsection in the main text, as the description of the model set-up is usually part of the main text for inversion articles.

2. The WFMD product reports the highest absolute XCH8 values among the three datasets and shows the largest bias relative to TCCON (7.5 ppb; see Table C2). While the WFMD XCH8 dataset shows moderate differences from both prior and posterior simulations compared to SRON and BLENDED, the fluxes derived from WFMD are consistently the lowest (Figure 10b). This discrepancy raises discussion here.

Could the lower flux estimates be a result of overestimated background subtraction? Furthermore, WFMD includes approximately 30% more retrievals than the other products. Could this may be due to less stringently filtered, allowing residual uncertainties to remain, which could contribute to the observed differences in flux estimates?

The lower emission estimates derived from the WFMD inversion are indeed a crucial element of analysis in this study. This element is discussed at the end of Section 3.3. We did not manage to identify a unique origin of these strong negative increments: the distribution of obs-sim is overall positive (Fig. 9) but with values closer to 0 than SRON, so we expect lighter increments than for SRON inversion – this is not the case. We investigated the background optimization (strong negative increments could have compensated strong positive increments on the background, which is not the case) but it was not conclusive either (Fig. D4, D5).

This is likely due to a from subtle balance within the inversion process: the differences in spatial distributions of background increments and emission increments with the other products could

come from differences in the separation of background and emissions when using WFMD observations. The high number of retrievals could also result in overfitting in the optimization process, and/or residual XCH4 uncertainties due to less stringent filtering. This topic has to be further explored to allow better understanding of the process.

The results of Rona Thompson presented at EGU 2025 also showed similar relative differences between products (Prior: 27.4 Tg/yr, SRON: 26.4 Tg/yr, BLENDED: 23.7 Tg/yr, WFMD: 19.0 Tg/yr, ICOS: 22.9 Tg/yr). These simulations were made with another model (FLEXPART). It cannot be considered a strict validation, but it suggests that WFMD observations do indicate lower CH4 emissions than the other products in Europe in 2019.

Reference: Thompson, R., Schneider, P., and Stebel, K.: Using different TROPOMI XCH4 retrieval products in atmospheric inversions of CH4: a comparison and reconciliation over Europe, EGU General Assembly 2025, Vienna, Austria, 27 Apr–2 May 2025, EGU25-9567, https://doi.org/10.5194/egusphere-egu25-9567, 2025.

3. While the study uses surface observations for the inversion, have the authors considered assimilating ground-based total column data, such as from TCCON or COCCON? These measurements are widely used to validate satellite products, as both observe total atmospheric columns and are therefore directly comparable. Incorporating such datasets could provide a more representative constraint on regional methane emissions and may help improve flux estimates, particularly in regions where surface observations are sparse or not fully representative.

The use of TCCON data in this study is limited to the comparison of XCH4 datasets and is presented in Annex C. We do not use the data for the comparison of the emissions estimated from the inversions, thus it is not used as a validation dataset. We use in-situ ground-based measurements for the validation (Sections 3.3, 3.5).

We have considered the assimilation of ground-based total columns from TCCON or COCCON. We decided to assimilate in situ measurements as an independent dataset because the 19 stations used in this study offer higher spatial coverage than the 5 (COCCON) + 7 (TCCON) sites available in 2019 (with 2 redundant sites in Karlsruhe and Sodankylä).

**Specific and technical comments:**

Line 126-127: the authors reference previous work to justify scaling the errors by a factor of 2, but a brief explanation here would enhance clarity for readers unfamiliar with that context.

Considerations on the definition of errors have been grouped in Section 2.1.3 (L.195-214). Specifically, we have described more thoroughly the error in the SRON product, and explain that the product authors « suggest to multiply them with a factor 2 to reflect the scatter of errors in the TCCON validation ». This is based on the ReadMe of the SRON product (see reference to Landgraf et al., 2024, page 10 of the document), but no further explanation is available.

Line 207: Figure 3 is referenced before Figure 2 in the text. The order of figures should be revised for logical flow.

The order of references has been reversed in the text (L. 175,176).

Line 207-208: The phrase "BLENDED lower concentrations" is unclear. Does this mean that only the lower end of BLENDED values agrees well with GOSAT? Consider clarifying this point. Additionally, including GOSAT's spatial distribution or mean value in Figure 3a would improve visualization and support the comparison.

The explanations have been clarified: the statement that BLENDED XCH₄ are lower than SRON and WFMD for all months is quantified (L. 184,185), and the statement that BLENDED is the closest to GOSAT is mentioned (« Only BLENDED aligns rather well with GOSAT, due to its correction of the TROPOMI-GOSAT bias »).

We chose not to compare averages of TROPOMI XCH4 with GOSAT XCH4 for the readability of the Figure. Also, the sampling is very different for both instruments: the comparison of averages can therefore deceive the reader in thinking than the differences are due to biases between products, whereas they can be caused by differences in coverage.

Figure 3: It seems the errors for SRON and BLENDED largely overlap in panels (b) and (c). If this is the case, please clarify it in the figure caption.

BLENDED errors are retrieved from SRON, so errors from both datasets are exactly equal. It is explained in the main text at L.129 and 196. The sentence « SRON and BLENDED errors are exactly similar, as BLENDED errors are directly retrieved from the SRON product. » has been added to the caption of Fig. 3 to improve readability.

Line 240: typical of SWIR retrievals >>> type?

This has been changed to « characteristic of SWIR retrievals ».

Line 408: The analysis indicates that differences are closely linked to across-track pixel index, aerosols, and SWIR albedo. While the latter two are physically intuitive, the role of across-track pixel index is less clear, as it represents only a positional identifier. Could these index be indirectly capturing information related to albedo, aerosol, or stripping artifacts in the retrievals? Further clarification would be beneficial.

The link between the positional across-track ID and striping patterns is described in L.342. A reminder « across-track pixel index (thus striping patterns) » has been added at the mentioned location (now L.365) to remind the reader of the link between the ID and striping effects.

Line 519: Background removal is known to be challenging. The manuscript would benefit from a brief explanation of how the background is defined and removed in this study.

The background mentioned here is composed of the lateral and top boundary conditions, the initial conditions and the stratosphere concentration field that are taken from the CAMS global product. It is defined in Section 2.4.2 and we have added here a reference to this section.

Line 531-532: result of >>> result from?

The correction has been done.

Line 560: The authors note poor improvement over Scandinavia. Could extending the study period improve coverage and reduce uncertainty through increased data availability? This point deserves further discussion.

Poor improvement over Scandinavia is due to the lower coverage in this region (Figure 1), in particular in winter (many pixels with no observations for JFM and OND in Figure D3).

The extension of the study has been considered to generalize the analyses and capture seasonal signals (only one year is not sufficient, some temporal variations could be specific to this year). It has not been implemented because of the already high number of simulations performed in this study, and their high computation cost. However, it is not clear that extending the study period would reduce uncertainty: the increased coverage would provide more data, but in the meantime the inversion would have more emission and background components to constrain, resulting in the same « constraint capacity » of the inversion. The ratio between the number of observations in Scandinavia and the one in other parts of the domain would remain approximately similar, thus the inversion would behave similarly and provide low constraints in Scandinavia.

The study of high latitude regions has to be the object of dedicated studies, which has already been the case using TROPOMI data: the studies of Tsuruta et al. (2023) and Lindqvist et al. (2024) are cited in the manuscript, see also the more recent study of Kivimäki et al. (2025).

Reference: Kivimäki, E., Aalto, T., Buchwitz, M., Luojus, K., Pulliainen, J., Rautiainen, K., Schneising, O., Sundström, A.-M., Tamminen, J., Tsuruta, A., and Lindqvist, H.: Environmental drivers constraining the seasonal variability of satellite-observed methane at Northern high latitudes, EGUsphere [preprint], https://doi.org/10.5194/egusphere-2025-249, 2025.

Appendix C: COCCON is increasingly used as a complementary dataset to TCCON, particularly in regions lacking TCCON coverage. The authors could also consider incorporating COCCON measurements to provide additional reference data for satellite validation, thereby enhancing the spatial representativeness of the evaluation.

See the answer to General comment #3: we agree that COCCON would provide additional data enhancing the comparison of TROPOMI XCH4 products. The use we make of TCCON data in this study is limited to the comparison of XCH4 datasets: as it is not directly used for the comparison of the emissions estimated from the inversions, it is not an evaluation of our results but more a complementary comparative description of XCH4 products. For this reason, the section was moved to Appendix and we did not consider using more data from COCCON. For further studies, it will be very beneficial to use both observational networks to enhance the spatial representativeness of the analyses.

---

## Author Comment (AC3)

**Author response - RC3**

We thank the reviewer for their thoughtful comments. We respond to them below: the comments are copied hereafter and shown in black, our author responses in blue; suggested new manuscript text is indicated in green. New line numbers in the revised manuscript are provided.

**General remarks:**

At the beginning I would like to underline that this article is definitely worth publishing as its message is of the most importance while satellite products like CH4 concentration distribution are becoming more and more popular and conclusions taken from those measurements have political and economic importance. Authors did extremely good job and provided variety of analyses regarding three basic products related to CH4 column concentrations. The main aim of this article is to answer the title question. The role of scientific article is to give evidences and logical arguments with statistical approach to set up the thesis and confirm it (or discard it).

The scientific level of this article is very high. Large part of the paper describes methodology applied to obtain particular products from TROPOMI instrument. Secondly, modelling of the methane fluxes was described. This was done to bring the final product, i.e. CH4 fluxes to the readers. From the point of view of scientist not working with satellites measurements the paper is very long and frequently refers to other publication describing in detail particular techniques and methodologies. In my opinion it resulted with a patchwork of information not deep enough to understand the products without reading those referred publications and making the storyline heavily disturbed. All this looks like authors wrote the paper and gave the title question at the end. I bet that readers of this article will be curious what is the answer to the question. And presentation of that problem should be a main goal of this paper. Or authors can change the title easily to definitely more scientific oriented, like "comparison of three products.....", where such structure of article will be consistent with the title.

This comment and some of the comments below are driven by the fact that the title of the manuscript was misleading. The title question indeed suggested that the question will be answered in the article, and we provide more of a comparison, rather than we provide an exhaustive answer to the question of the consistency of emissions from the inversions. Here, we compare the products and the emissions estimated from inversions, and try to investigate the drivers of the differences that we see.

The title is revised to "Assessment of the differences in European CH4 emission estimates from three TROPOMI products". With this title, the structure of the article is more logical: we first compare directly the products (XCH4, coverage, errors, vertical profiles) in Section 2, then look for the drivers of these differences in Section 3.1 and continue the comparison in the next sections: obs-sim distributions in Section 3.2, emissions derived from inversions in Section 3.3, evaluation with surface stations in Section 3.5, with another look to the drivers of differences (for emissions this time) in Section 3.4. We investigate when the products are consistent or not, but do not provide a definitive answer to the previous title question, which has been changed.

The consistence of flux/emission product should be thoroughly analysed with detailed prepared tools. This is where OSSEs helps a lot and article clearly benefits from this approach. Consistence is also expressed in statistical way, and here the maps are extremely helpful. However to answer the question authors should present a clear and trace line leading from methodology to

consistency at pixel, region, country and continental scale. Here, analysis is dissolved in many technical references which are not relevant to the final answer.

We have modified the manuscript to clarify the organization of Section 2 (see answers to specific comments), make Sections 1 and 2 more concise. We have also tried to highlight a clearer structure, introducing it more carefully in the abstract and introduction, and bridging with the conclusion. Content that was presented in Section 2 and not used in Section 3 has been removed, and we have removed the subjective statements.

I see clear lack of hypothesis verification and many subjective statements not explaining if we can or cant obtain clear consistency of 3 TROPOMI products. My expectation for this article would be a detailed discussion what affects consistency of the 3 outputs in terms of methane flux distribution (temporal and special).

Discussion of similarities and differences between the products. Those only which are relevant to final conclusions.

Discussion of metrics applied to judge consistency of 3 products or any pair of those.

Discussion of results in form of the prove of the hypothesis with detailed description of the discrepancies and similarities of the product results (final ch4 fluxes)

Discussion of possible errors and uncertainties of the fluxes and its relation to consistency.

If authors find it necessary comparison with ground-based observations which in my opinion is irrelevant to the question (in title).

One would expect that the test of consistency can be done using known emission distributions for methane (large point source and large area sources). There are some verified sources of methane which can be potentially used for such purpose. However, again, agreement with real data is not matter of product consistency but rather reliability. Authors should look at the comparison of the products and condition when they deviate from consistency. That would be an answer to the title. In the form they proceeded it's a mess.

4D-Var inversions have been performed in this study, and therefore we do not have direct access to posterior uncertainties. This lack of uncertainty range intrinsically limits the analysis of consistency, as we cannot statistically check if estimated emissions are consistent or not. For this reason and to clarify the structure, we have modified the title of the article.

However, the manuscript is a discussion of similarities and differences between the products, at different steps of the inversion process: observations, comparison simulations vs. observations, derived emissions. At every steps, the distributions are compared in terms of average, deviations, spatial and temporal variability. When differences are observed, they are investigated (Sections 3.1, 3.4) through detailed analysis with metrics: SHAP values to evaluate feature contributions to XCH4 differences in Section 3.1, relative increments in the case of OSSEs in Section 3.4.

The objective of this study is to assess whether TROPOMI data users would find estimated emissions that are consistent if they perform inversions. In this case, we evaluate « consistency » with respect to the aim of building emission inventories at the pixel, country and continental scale. In this perspective, results can be compared to the ranges of estimates of top-down budgets: 31 [24–36] TgCH4/yr for Europe (not exactly the same mask as in this study) for 2010-2019 in Saunois et al. (2025), and 19 [15–23] TgCH4/yr for EU27+UK for 2009-2021 in Petrescu et al. (2024), cited in the manuscript. We have clarified the way we consider the consistency in the conclusion, but did not include the comparison of the ranges as the differences in covered areas and periods could be misleading for the reader.

Also, the uncertainty reduction (independent of the observation vector) can be estimated through OSSEs: computing the standard deviation of priors and the one of posterior emissions across the ensemble samples, the ratio of the two gives the uncertainty reduction. For the total budgets, it is estimated to 78% reduction for SRON and BLENDED, 74% for WFMD and 51% for surface stations. However, we do not use these values for setting error bars to our budget estimates, for two reasons: 1) the poor statistics of this ensemble (4 samples) and 2) the lack of proper uncertainty estimation for the prior as it is not provided with the emission inventories.

**Specific comments:**

**Abstract.**

Abstract should be a kind of teaser for what reader will find inside the article. It is. However, the expectation of answering the question is not confirmed in it. One can have feeling that the article will contain a chaos of information what indeed is true.

The obvious comment that XCH4 differences are related to aerosol and albedo one can find in any of publication related to any of those products. Authors should be a bit more constructive here giving something new in that field (e.g. quantification of those effects).

We quantified the impact of these parameters on the XCH4 differences between pairs of products. We have evaluated the impact of XCH4 differences on the emissions through the *Diff* OSSE scenarios, but did not performed separate analysis for albedo and aerosols. It would have required to assimilate bias-corrected products in inversions, and thus modify the XCH4 distributions. This is an interesting perspective and we mention it as future work in the Conclusion.

However, it was chosen not to perform this analysis in this study for three reasons. First, there is already a lot of content in the article. Secondly, the only difference between SRON and BLENDED is the XCH4 bias correction of the effect of albedo, aerosols, striping, etc. in BLENDED. The small differences in the results of inversions suggest that these parameters do not impact that much the derived emissions (this statement is qualitative, its quantification is required in future work). Finally, we have chosen to focus on parameters that do not require to modify the XCH4 distributions, in order to assimilate products as close as any data user would do, with the perspective of application uses. We agree on the fact that the quantification of the albedo and aerosol impact on the emissions is an important next step.

Comprehensive abstract should not contain contradictory statements : e.g.

"European CH4 emission budgets show relative increase of 2%, -1%, 33%..." - whatever relative increase is, its clear products are not consistent. Or they are, because 33% in compare to 2% is not much?

"Seasonal emissions are highly correlated..." – so, probably we will look for bias in products. So, if the correlation is a measure of consistency the answer is yes, if bias – the answer is no.

"Using consistent error definition....for more consistent emission estimates" – what is consistent error definition at this stage its hard to understand but it suggest that it will play a key role in the answer. But it is only "....paving the way..." so most probably the answer is again: No.

Article will be much more understandable if authors will focus on main message which is the answer to the title question and will build the well structured hypothesis verification.

See the answer to general comments regarding the title. The abstract has be modified accordingly to clarify the comparison and make the trace line of the article clearer.

**Introduction.**

First few sentences (lines 17 - 24) are so much obvious and frequently repeated. Its now primary school level knowledge. I think this article is not giving any new concept about those topics and as being submitted to ACP, every person who will read it will be aware of those basic facts about methane.

Next few sentences (lines 25 - 29) are referring to BU, which is later not compared with nor used in any form. Authors also do not provide any input to this method of methane budgeting. Critical

approach to the BU EU methane budget is not relevant if there is no better proposition from authors side.

We agree with the reviewer that there is no contribution to these topics from this study. The first two paragraphs have been grouped and shortened. A few sentences were kept to introduce notations and provide context for why it is important to have robust TD budgets.

Definitely there is a lack of introduction about satellites, especially those used for XCH4 measurements. There is a whole constellation of such platforms nowadays and role of S5P in that land-scape would make a good background for this paper. Please, describe or list all products currently used for XCH4 retrieval. This will help to understand the problem of coherence of product.

A more detailed list of the currently used instruments for XCH4 retrievals has been added to the introduction (L.28-33). The role of S5P is detailed when mentioning the resolution and coverage, as well as with the use of TROPOMI data for various types of emission quantification.

Secondly, please discuss where the products were applied (examples of application of those products – like lines 40 and 41, but with deeper explanation of problem in term of coherence of those products). How much those products differs for O&G facilities, mines, landfills...

The use of several products in a single study is rather limited and described in L.58-64. The of the consistency of the products is not analysed in the studies listed for O&G facilities, mines, landfills, etc. This is what we want to adress in this study.

Be specific while mentioning some aspects of work:

Line 48 – prior vertical profile of what parameters?

The mention « prior profile of CH4 mixing ratios » has been added.

Line 49 – stratospheric background of what?, aeroslos... - do not leave "..." here. Discussion of those effects are essential for this paper and should be introduced carefully in that chapter (instead of GHG effect).

The « ... » has been removed and background has been described. The mentioned effects are identified through the ML analysis as drivers of the XCH4 differences between products and are detailed in Section 3.1, but they are not studied in inversions (see previous answer to the comment in the abstract).

Line 49 – what retrieval?

The formulation was not clear, it has been changed.

Line 50 – scattering of what? What spectral range? What issues are mentioned here?

The sentence has been changed to « ... scattering due to the presence of aerosols ».

Line 52 - ...routinely updated. – provide the reference.

To be more accurate, « routinely » was changed to « iteratively ». The iterative versions are described in the provided references.

Line 54 – avoid slang, like "beta research product".

The change has been done, « beta » has been removed.

Line 54 – At this stage destriping algorithm might be at least briefly explained.

A brief description of the principle of this algorithm has been added: « a destriping algorithm, based on moving median smoothing in the across-track directions and flight directions, has been developed for new XCH4 data ».

Line 67 - ...the consistency and applicability... those terms require the measure (scale) would be nice to have few words in introduction about it.

This sentence has been removed for concision.

Please explain why Europe is a targeted area for this article and why 2019 was chosen. It is important because later authors discuss a seasonal effect, what in relation to single year is not usual. Especially that there is a definitely longer period available now (2025). Effects in that year might not be representative for a "usual" seasonal singals.

As mentioned at L.69-70, the study focuses on Europe due to its extensive network of surface stations, as we use them to support the comparison. Other regions (e.g., North America) could have been chosen, but we wanted to build on the studies focusing on Europe that have been published in the last decade.

We chose to focus on 2019 to have a full year available (no TROPOMI data before april 2018) and to avoid patterns of emissions related to the lockdowns (2020 and 2021). The extension of the study has been considered to generalize the analyses and capture seasonal signals. It has not yet been implemented because of the high computation cost of the performed simulations. A follow-up study should inquire the long term implications of using a given product rather than the other.

**Data and methods**

Here, my opinion is that authors can easily shorten this long chapter giving a list of good references to each product. Only aspects important for a latter discussions should remain.

This section has been restructured and many subsections have been shortened for concision, and to ensure we focus on elements that are useful for the analyses of Section 3.

Lines 100 – 108 – are those details important for the products coherence?

This paragraph recalls a few parameters of the TROPOMI instrument and the error requirements, and also serves as an introduction to the description of products. Some unnecessary details have been cut for concision.

Line 115 – This regularisation is more frequent referred as Tikhonov, or Tikhonov-Philips. Again level of description is strange. We cant see any reference to this algorithm anywhere later. Line 116 - ...state vector... not explained what it is and how it may affect coherence? Line 119 – VIIRS instrument is certainly important but does use of it influence the coherence of the products?

These sentences have been removed for concision, and because they do not provide useful information for our analyses.

Line 123 – ...(qa\_value... later in line 137, and variable xch4...in line 152) is such variable need to be mentioned here? If yes please describe what it is.

For the 3 products, the names of the dedicated variables were removed, we use « quality flag » (L.108, 125, 134).

Line 124 – why do we need to know that there is 4731034 observations – how this number affects the conclusions? Table 1. contains this information.

To avoid repetition with Table 1, this sentence was removed.

Line 124 – if destriping is applied to only "new" data and it is important for coherence in 2019 – why authors didn't insist to apply it also for 2019?

We identified striping as one of the main drivers of XCH4 differences. However, the destriped data is not available in 2019. Data could have been reprocessed, but at the cost of modifying XCH4 data. In the perspective of using data « as an user would find », we decided not to reprocess the XCH4 observations with an additional algorithm.

Line 126 – what is the single sounding precision?

My comment to authors regarding uncertainty calculations: It is a crucial part of this paper and authors underline it the conclusions. Why it is so poorly explained in the chapter 2. Please devote some space to explain directly how you apply the uncertainty chain in your calculations. Give a table with some values.

Considerations on the definition of errors have been grouped in Section 2.1.3 (L.195-214), to make the structure of the article clearer. The terms such as « single sounding precision », « pseudo noise »... have been changed to clearer words. We have summed up the definitions of errors provided in the ATBD of each product (SRON/BLENDED on the one hand, WFMD on the other hand). In both products, the noise in the measured radiances is propagated in the retrieval algorithm. Then, the post-processing of this propagated error is based on the comparison of  $XCH_4$  observations to TCCON: to reflect the scatter of errors in this TCCON validation, different post-processing are implemented across products.

- For SRON/BLENDED, the authors suggest to rescale the provided error by a factor of 2.
- For WFMD, the provided error already take into account a linear rescaling (Eq. 1).

The difference in the individual errors provided by the products is what we want to highlight here.

Line 133 – TCCON observations bias regarding GOSAT not TROPOMI product is not referenced. Explain why do we need this correction and how does it affects the consistency of products?

The removal of this bias is made by Parker et al. (2020), which provide the product. It is applied to the GOSAT observations used to create the BLENDED product (Balasus et al., 2023). The effect on the consistency of products is described further (Section 2.1.3), when comparing the  $XCH_4$  distributions.

Wouldn't be good if TCCON is used for product consistency check?

A comparison of TROPOMI XCH4 products to TCCON is provided in Annex C. Initially in the main body of the text, it was moved to Annex as it is not directly used for the comparison of emissions from the inversions. It provides a comparison to independent data, interesting when comparing the products, but does not serve directly the main message of the article.

Line 136 – limitations are extremely important for this discussion – no references, no comments?

These limitations are discussed later (Section 3.1), so we do not insist on them at this specific location of the manuscript.

Line 137 – ga-value – what product it refers to? SRON?

See the answer to the previous comment on quality flags variables.

Line 137 - ...recommended by the authors. what is rationale behind that? Authors of what? Line 141 - ...provided errors... please give a reference and describe more specifically. Line 158 – Section 2.1.3. contains very poor description referring partly to section 2.1.1.! And defines as st dev of retrieval noise – not explaining what it is and how does it is calculated. In terms of XCH4 values. Please, provide this information somewhere. Also pseudo noise should be defined if it is relevant to final conclusions.

See the answer to the previous comment on errors: the description of the error definition for each product has been restructured in Section 2.1.3.

Equation 1 – no variable symbol definition is provided.

Equation 1 and other considerations on the definition of errors have been grouped in Section 2.1.3 (L.195-214). Sigma and sigma\_hat are now defined in the text (L.205 and 206).

Line 169 - ...is relatively mature product, even though sparse. - If it is important for BLENDED product please indicate whet does it mean.

This subjective sentence was removed for concision.

Line 174 - 9.2 ppb bias is repeated?

This information is mentioned twice: one in the BLENDED description when mentionning GO-SAT, and one in the GOSAT description when making the link with BLENDED.

Line 194 – what scene description variables are important here?

The most important ones have been added (cloud fraction and SZA), the other ones can be found in the added reference (ReadMe of the SRON product).

Line 195 – Spatial patterns of differences.... - Of what?

The sentence has been modified to « of coverage differences ».

Line 199 - ...change of pixel size. – why this year was chosen not e.g 2020?

2019 was chosen to check if the change of pixel size has a major impact on coverage (which is not the case), and to avoid emission patterns related to the various lockdowns in Europe (2020 and 2021).

Line 203 – If CFC was exceptional, maybe longer period of comparison (2019 – 2024) would be beneficial to cancel such effect.

The extension of the study period was considered but not implemented due to its high computational cost. Moreover, the variation of coverage due to high CFC is not detrimental for the comparison of emission estimates. Further studies will apply multi-year inversions to assess this impact, beyond the scope of the present paper.

Line 205 – where is a Figure S1?

Figure S1 is available in the Supplementary material. A mention « in the Supplement » has been added for clarification.

Line 206 – there are statistical tests for non parametric hypotheses? What does "approximately" mean here?

This subjective statement was removed, as the gaussian aspect (or not) of the distribution is not used later in the analysis.

Line 209 – 9.2 ppb mention 3rd time

This repetition was removed.

Line 210 – why there is no such comparison at the Figure? Its not clear how the GOSAT average is calculated if there is so big difference between the pixel size and number of data?

The comparison to GOSAT averages was removed, as the large differences in coverage between TROPOMI and GOSAT products makes the comparison of averages not relevant.

Line 213 – There are over statistics describing distributions not only average (expected value).

Line 220 – how does offset is calculated (difference of averaged monthly values?), why not calculate it for each products pair.

Line 220 – what does approximately means - why not to give st dev?

The differences of averaged monthly values have been quantified (average and standard deviation) for each pair of products (L.184-185), and the word « offset » was removed as it was confusing.

Line 230 – what noise authors refer to?

Line 232 – what is "pseudo noise"?

Line 234 – 3.9 is not order of magnitude lower than 12.2

Line 234 – scaling up the "error" is unusual procedure that requires additional clear evaluation – please provide the reasons behind this transformation. Being smaller is not one.

Line 235 – Author should explain why Eq1 is applied and where does those coefficients come from.

Line 236 – no definition of error is given (anywhere), how can be error understood here: uncertainty, systematic error (bias),

Line 237 – also here observational error is without definition...please provide.

See previous answers to comments on the error definition: we have grouped all the error considerations of Section 2 in L.195-214. We replaced « order of magnitude » by « factor 3 ». We provide the explanation of the error scaling in WFMD (and the corresponding references), and why we apply a similar scaling to SRON/BLENDED errors.

Line 244 – Huge difference of XCH4 (200ppb) is hardy evaluated by fig 4. Can authors provide a panels with a view on higher pressure levels so this, most important part of profile is clearly visible?

Figure 4 has been modified to provide a zoomed view of the high-pressure levels.

Line 275 – please give a reference (...less than 2 weeks)

A reference to a study on the transport of air masses in Europe (Nygård et al., 2023) has been added to justify this statement.

A whole chapter 2.3. should be target at application of products, its importance for emission estimation and coherence of the emission patterns. Details can be linked to literature and equation 2 is not necessary, or authors should refer latter to it discussing the sensitivity of model to pressure profile in particular products if different.

Chapter 2.3 was shortened for concision, Equation 2 was removed as it is not refered to later.

2.4. chapter is again basing on many references and construction of inversion model principle is not relevant if not used later in results chapter. This article is not basing on particular matrices

used in eq.3. If statement in line 231 can be explained without introduction of the Eq 3 than authors may revise need of that equation and chapter 2.4.1 and large part of 2.4.2.

Sections 24.1 and 2.4.2 were restructured to make them more concise and remove unnecessary elements. However, we believe that it is worth quickly restating the principle of our inverse simulations, as well as describing thoroughly the set-up to ensure replicability and clear statement of the hypotheses.

Line 305 and 306 – why usually 20 – 30, what is the cost function, is it relevant for this discussion?

This confusing statement has been removed. The idea was to indicate that the 95% reduction criteria is in practice reached after 20 to 30 iterations. But this is not useful for further analyses.

Line 315 - is it RSD? Why set it to 100% ? How does it correspond to "error" definition in model? Line 318 – why 2% not 5%

The mention « relative » has been added for the standard deviation. The values have been chosen based on commonly shared values, in particular from the intercomparison of CH4 inversions in Europe (loannidis et al., 2025). The study of Szénasi et al. (2021) also helped chosing these values. The background error was decreased from 10% to 2% to ensure that the weights of corrections on emissions and background are not too unbalanced.

Reference: Ioannidis, E., Meesters, A., Steiner, M., Brunner, D., Reum, F., Pison, I., Berchet, A., Thompson, R., Sollum, E., Koch, F.-T., Gerbig, C., Wang, F., Maksyutov, S., Tsuruta, A., Tenkanen, M., Aalto, T., Monteil, G., Lin, H., Ren, G., Scholze, M., and Houweling, S.: An inter-comparison of inverse models for estimating European CH4 emissions, Earth Syst. Sci. Data Discuss. [preprint], https://doi.org/10.5194/essd-2025-235, in review, 2025.

Line 319 – What does "large" means in reference to model error?

The criterion used for defining « large » differences is > 100 ppb. This is an arbitrary choice to avoid the inversion to focus on very large obs-sim differences.

Line 325 – what is "poor" quantitatively? How does it affect XCH4?

This subjective statement has been removed.

Line 331 – Confirmation is not supported by logic and statistics, model can run wrongly in some areas and this will also result as same distribution of removed observations in all products.

This statement was indeed not well supported, it has been removed.

**Results**

I personally found this chapter very interesting and well prepared. It contains a lot of earlier briefly or poorly explained information and references. It looks like it was written by different person or group than chapters 1 and 2. My only general comment in relation to the results is to focus on coherence of products (like fig.7). Give as much quantitative comparisons and avoid subjective statements like "could be related", "possibly" etc.

We have taken this comment into account and have tried to rephrase as much as possible these expressions, while providing quantifications. However, some occurences have been kept to insist on the fact that we suggest a explanation, but cannot certify a unique cause.

Chapter 3.3 - Seasonal cycle and annual budget are the parameters which are affected by boundary conditions and exceptional situations if they are referred to a single year period. In my opinion authors could make the analyses longer for this comparison.

The expression « seasonal cycle » has been changed to « temporal variations »: as mentioned, seasonal cycle cannot be analysed through a 1-year analysis. See previous comments on the extension of the study period on this aspect.

Line 506 – Figure D5 is appearing before Fig D4 is mentioned (line 520)

The order of these figures has been changed.

Line 612 – There is no information what is uncertainty of calculated yearly emissions and it is hard to judge if change from 25.2 to 25.3 is statistically significant. Definitively term "slight" is not quantitative.

See the answer to the general comment for uncertainty considerations. The subjective word « slight » has been removed.

**Conclusions**

I will repeat my general remark here. I would like to know the answer to the title question. If it hasn't been given in chapter 3, I expected it to appear here.

See answer to the general comment.

Line 654 – what does the "holds great promise" refers to – list a reference or be more specific.

This expression has been removed and replaced by a reference to the review of current and scheduled satellite observations from Jacob et al., 2022.

Line 658 – Why consistency of products is a crucial for interpretability? Why does one product cant be just better than other two?

This sentence has been rephrased to « essential to maintain the comparability of the subsequent analyses based on these products »: indeed, consistency is not crucial for interpretability. But as the products are already used for studies and publications, it is important to know how we can compare these studies.

Line 660 – if the lower quality of data would be chosen – coverage factor would be greater, isn't it like that? This both parameters are basically inverse dependent. Please give a measure that change of data quality filter affects the emission distribution. It wasn't in the result, beside the counterintuitive statement that larger errors results in enhancements of the emissions.

The « high-quality » mention has been removed. In this study, the impact of quality filtering was quantified through the impact of coverage in the « Common » OSSEs. Based on our results, we can only provide a measure for these OSSEs (in terms of relative increment), but not on the real emission distribution.

Line 661 – what deeper understanding refers to? Please, provide the quantities for each factor in which it impacts the retrieved fluxes:

Quality filtering,

Spatial coverage,

Spatial distribution,

Uncertainty of observation,

Ability to constrain emission (not discussed earlier).

This has been discussed in the Sections 3.3 and 3.4, but cannot be summed up with a single quantification. In Section 3.4, the metrics is the relative increment and enables to quantify the impact of varying coverage and observation uncertainty on the resulting emission distribution, in terms of spatial and temporal constraint. The impact of differences in the distribution of XCH4 has been discussed with the *Diff* OSSEs. It is rather a qualitative analysis than an absolute comparison of the impact of the mentioned factors on estimated emissions.

Line 665 – Application of ML to discover that retrievals are dependent on aerosol, albedo and striping is not new – all that was clearly reported in cited literature. Why it is not an object of this investigation? It is what I expected.

See previous answer to the Abstract specific comment. The ML application enabled to quantify the impact of these parameters on the differences of XCH4 between products, which had not been done before (it had been done by Balasus et al., 2023, but in the case of the TROPOMI-GOSAT difference). The quantification of the impact of albedo, aerosols and striping on emissions is a very important perspective. We chose not to focus on these parameters in the OSSEs, as it would have required to modify the XCH4 products (providing a correction). We wanted to use the products « as an user would do ». The perspective of the albedo/aerosols/striping impact on emissions has been highlighted as the most important future work.

Line 666 – Recent.... – why not to calculate how much those activities (developments) can improve the emissions coherence (this would be a great answer to the question).

This would be a very interesting study. However, it would require a detailed study comparing the previous versions of the products, that are not used anymore. It would not enable us to make predictions on future improvements. We prefered to give a snapshot comparison: a comparison of emissions estimated from inversions given the products available at the moment. Comparing the impact of the improvements goes beyond this study. The statement on the fact that future improvements should « improve the consistency of products » has been removed, as it is not supported by a result of this article.

Line 681 (probably most important one) – Without the uncertainties this numbers are not giving proper answer to title question. Please set the hypothesis and confirm or negate it with proper statistical analysis.

See the answer to general comments on the uncertainty considerations.